# Direct observation of subunit rotation during DNA strand exchange by serine recombinases

Gillian M. Cadden [1,2], Jan-Gero Schloetel [1], Grant McKenzie [1], Martin R. Boocock[1], Steven W. Magennis [2] ✉ & W. Marshall Stark [1] ✉

Serine recombinases are proposed to catalyse site-specific recombination by a unique mechanism called subunit rotation. Cutting and rejoining DNA occurs within an intermediate synaptic complex comprising a recombinase tetramer bound to two DNA sites. After double-strand cleavage at both sites, one half of the complex rotates 180° relative to the other, before re-ligation of the DNA ends. We used single-molecule FRET (smFRET) methods to provide compelling direct physical evidence for subunit rotation by recombinases Tn3 resolvase and Sin. Synaptic complexes containing fluorescently labelled DNA show FRET fluctuations consistent with the subunit rotation model. FRET changes were associated with the rotation steps, on a timescale of 0.4–1.1 s$^{-1}$, as well as opening and closing of the gap between the scissile phosphates during cleavage and ligation. Multiple rounds of recombination were observed within the ~25 s observation period, including frequent consecutive rotation events in the cleaved-DNA state without evidence of intermediate ligation.

Site-specific recombination defines a large group of natural processes in which DNA molecules are rearranged by breaking and rejoining the DNA strands at pairs of specific sequences ('sites'). Site-specific recombination can be very efficient and highly specific; as a result, it has many current and projected applications in biotechnology and synthetic biology[1].

The enzymes that promote site-specific recombination are called recombinases; of these, the serine recombinases make up a large group related by similarities of structure and mechanism[2]. The so-called small serine recombinases are proteins of about 200 amino acids (aa) comprising an N-terminal catalytic domain (~150 aa) and a C-terminal DNA-binding domain (~50 aa) joined by a short flexible linker. The structures and mechanisms of exemplar small serine recombinases have been studied in detail for many years, and a number of high-resolution structures of the proteins and their complexes (including reaction intermediates) have been solved[3,4]. The modular structure of these enzymes has led to the engineering of 'designer' recombinases that can promote recombination at a wide variety of target sequences[5–7].

Early analysis of the small serine recombinase mechanism followed the changes in DNA topology when supercoiled plasmid substrates were recombined. These studies, along with further biochemical evidence, led to the proposal that strand exchange (i.e. the breaking, rearrangement and rejoining of the DNA strands) was by a mechanism called subunit rotation (Fig. 1A)[8]. In this model, recombinase subunits bind to their partially palindromic 'crossover sites' as dimers, and two DNA-dimer complexes assemble to form a synaptic complex with the DNA situated on the outside of a recombinase tetramer[9–12]. Both DNA strands at the centre of each crossover site are then cleaved by attack of the recombinase active site serine residue hydroxyl groups on specific phosphodiesters, resulting in 2-nt 3′-OH overhangs and covalent attachment of the recombinase subunits to the recessed 5′ ends via a phosphoseryl linkage[13]. In the critical exchange step, one entire half of the cleaved synaptic complex (two recombinase subunits with their attached DNA half-sites) is proposed to rotate 180° relative to the other half, and the DNA ends are then rejoined by reversal of the cleavage mechanism. Wild-type small serine recombinases form synaptic complexes with specific DNA

[1]School of Molecular Biosciences, University of Glasgow, Bower Building, University Avenue, Glasgow, UK. [2]School of Chemistry, University of Glasgow, Joseph Black Building, University Avenue, Glasgow, UK. ✉e-mail: steven.magennis@glasgow.ac.uk; marshall.stark@glasgow.ac.uk

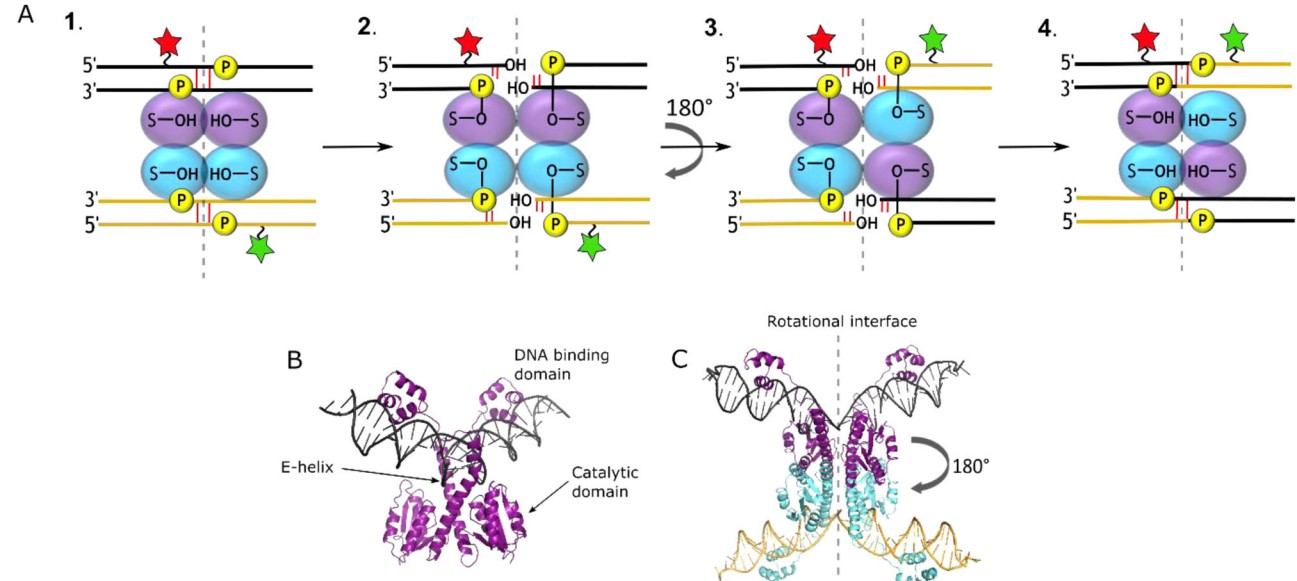

**Fig. 1 | Subunit rotation model of serine recombinase-catalysed site-specific recombination. A** 1. The synaptic complex contains a serine recombinase tetramer (purple and blue ovals) bridging two double-stranded recognition sites (black and yellow lines). The catalytic serine residue is shown as S-OH, and the scissile phosphodiesters are labelled P. Donor and acceptor fluorophores at DNA attachment points that allow the mechanism to be tracked by changes in FRET efficiency are represented as green and red stars. 2. Nucleophilic attack by the hydroxyl groups of the serine residues cleaves the double-stranded DNA with a 2-bp stagger, leaving ends with a 5'-phosphoserine linkage and a 3'-OH. 3. For strand exchange, one complete half of the complex is rotated by 180° using the flat hydrophobic interface as a rotational bearing. 4. Subsequent ligation of the strands takes place by attack on the 5'-phosphoseryl linkages by the deoxyribose 3'-OH ends. **B** Co-crystal structure of a wild-type γδ resolvase dimer bound to site I DNA. (PDBid 1gdt[7]) **C.** Activated γδ resolvase mutant tetramer bound to two site Is in the post-cleavage covalent protein-DNA synaptic intermediate state (PDBid 1zr4[9]).

architectures in plasmid substrates, such that right-handed rotation is favoured by loss of supercoils, leading to specific product topologies that are observed experimentally[3,4]. Product topologies consistent with multiple rounds of rotational strand exchange have also been observed[14–17], and other biochemical studies of the mechanism are also consistent with the model[2–4].

When subunit rotation was first proposed, the required rotation step was seen by some as structurally implausible, as it could lead to catastrophic dissociation of the two rotating halves of the synaptic complex. Alternative mechanisms that could give the observed product topology outcomes without rotation were advanced[3,18,19], though these failed to account for all the experimental data. However, the crystal structures of a cleaved synaptic complex of a γδ resolvase tetramer with its crossover sites provided strong supporting evidence for the rotation mechanism (Fig. 1C). In these structures, the recombinase dimers that are predicted to rotate relative to one another are held together by a flat hydrophobic interface, which potentially provides a smooth bearing for resolvase-DNA rotation, while extensive hydrophobic interactions are predicted to prevent synapse dissociation[9,20]. More recent structures provide further support for the subunit rotation model, showing recombinase tetramers with multiple 'rotational states' (i.e. relationship of the two 'rotating dimers')[21–23].

Direct observation of subunit rotation requires the implementation of single-molecule methods. Such techniques overcome the limitations of ensemble averaging and allow biological processes that are asynchronous to be tracked in real-time, and their rates measured directly. Although tethered particle motion (TPM)[24] and magnetic tweezers[25,26] techniques have been used to observe reactions promoted by serine recombinases, these methods do not report directly on structural transformations within the synaptic complex itself. Also, the mechanism might be affected by experimental factors such as applied torque and bead size[27]. In TPM studies of φC31 integrase-mediated recombination[24], various association and dissociation rates were determined, and behaviour associated with a 'gated rotation'

model was noted. However, the kinetics could not be directly correlated with nanometre-scale movements within individual synapses due to limited spatial resolution, so rates corresponding to individual steps of the mechanism remained undefined. Here, we directly observe the DNA rearrangements associated with the subunit rotation mechanism by observing intermediate complexes (both free in solution and immobilized to a surface) using single-molecule Förster resonance energy transfer (smFRET).

## Results

### Substrate design

We studied the mechanism of strand exchange by two small serine recombinases, Sin resolvase and Tn3 resolvase. In both cases, we used an activated mutant enzyme that is independent of the accessory DNA sequences, accessory protein subunits, and DNA supercoiling required for the wild-type recombinase activity, and which therefore promotes recombination between short (~30 bp) double-stranded oligonucleotide sites corresponding to site I of the natural *res* recombination sites[21,28].

Donor and acceptor fluorophores were attached to T bases near the centre of the site I DNA. The best positions were considered to be those with the smallest initial distance between the donor and acceptor fluorophores (which were therefore predicted to give large FRET efficiency changes as the cleaved DNA segments move during strand exchange), and, additionally, those which least inhibited the recombination reaction (Supplementary Figs. 1, 2; Supplementary Table 1, 2). In our subsequent experiments, we attached the fluorophores at the positions highlighted in Fig. 2A and B. Our 'linked site' DNA substrates (referred to as SUL25 for Sin Q115R and UL25 for Tn3 NM) for smFRET consisted of two site I sequences connected by a single-stranded polythymidine linker, with a donor fluorophore attached to one site and an acceptor on the other (Fig. 2A and B; Supplementary Fig. 3). The substrates were designed to favour the inversion reaction pathway and disfavour the alternative deletion pathway (Fig. 2C), which was

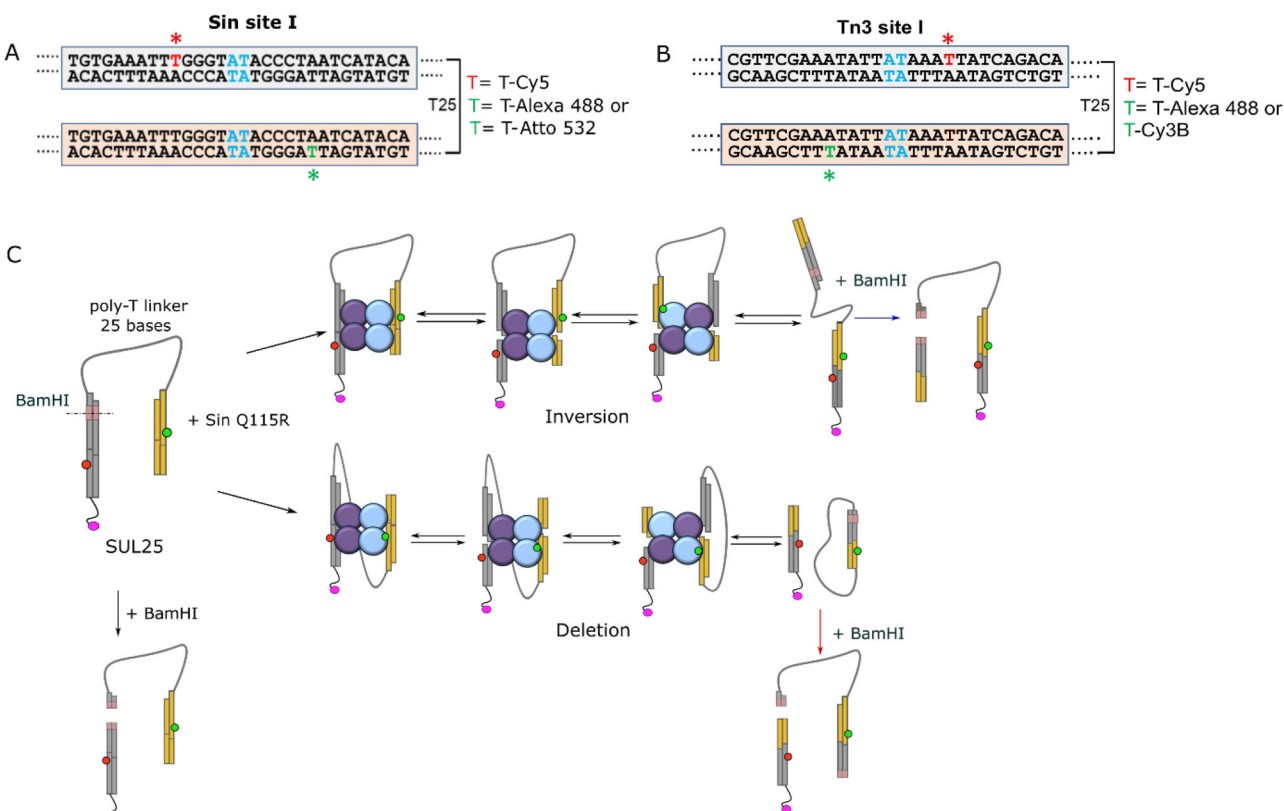

**Fig. 2 | Linked-site recombination substrates. A** Locations of the FRET donor (Alexa 488 or Atto 532) and acceptor (Cy5) positions in the linked-site Sin substrate SUL25. **B** Locations of the FRET donor (Alexa 488 or Cy3B) and acceptor (Cy5) in the linked-site Tn3 substrate UL25. **C** Potential recombination pathways for linked-site substrates. The dsDNA recombination sites are shown in grey and yellow and are joined by a flexible ssDNA linker (grey line). Resolvase subunits are shown as purple and blue circles and the biotin label attached to the 5′-end of the substrate is shown as a smaller magenta oval. Inversion (upper pathway) can be tracked by following distinct FRET changes at each stage. Deletion (lower pathway) is predicted to proceed with only small FRET changes.

achieved by restricting the length of the linker (Supplementary Fig. 4) so that the sites are constrained to be predominantly aligned as shown in Fig. 2A and B (see 'Methods' for further details)[10]. Note that in our initial experiments involving recombination of two linear site I substrates (Supplementary Fig. 2A) we found that recombination was more efficient for Sin Q115R at 37 °C (Supplementary Fig. 2B, C). However, this was not the case for Tn3 NM resolvase (Supplementary Fig. 5), and therefore we conducted all further experiments with Tn3 NM resolvase at room temperature. We varied the temperature for subsequent single-molecule experiments with Sin Q115R to investigate further (see below).

The inversion pathway is expected to cause a large increase in FRET as the fluorophore-bearing half-sites move together and are joined to form a recombinant site, whereas the deletion pathway is not expected to cause substantial FRET changes, as the fluorophores remain on separate sites in the recombination product. However, for both pathways, we predict a small but observable increase in FRET efficiency on synapsis of the fluorophore-bearing sites prior to strand exchange.

### Accessible volume (AV) simulations: modelling FRET pair distances and signal intensities

AV simulations were conducted to estimate FRET pair distances at each stage of the recombination pathway (Fig. 3). For the cleaved non-recombinant (cnr) and cleaved recombinant (cr) structure simulations, we used the crystal structure of γδ resolvase covalently linked to two cleaved DNA sites (PDBid 1zr4)[9], the only difference being the expected fluorophore positions before and after 180° rotation from the original 1ZR4 crystallographic configuration. We used the crystal structure for a γδ resolvase dimer bound to site I DNA (PDBid 1gdt)[7] to simulate the FRET distances for the final ligated recombinant (lr) structure. As yet, there are no high-resolution crystal structures for ligated Tn3 or Sin resolvase synapses. We used the low-resolution structural model of the uncleaved Tn3 NM resolvase synapse[11], which is constructed by juxtaposition of two 1GDT γδ resolvase dimer-site I structural units such that the separation and crossing angle of the two site I DNAs are consistent with small-angle scattering data.

### Recombination of freely diffusing complexes

Products of Sin Q115R-catalysed SUL25 recombination reactions were analysed by SDS-PAGE to provide an initial assessment of recombination and FRET efficiency. Each recombination pathway (deletion, inversion, or intermolecular fusion) generates products with unique mobilities on SDS-PAGE gels after restriction enzyme digestion at specific positions within the substrate (Fig. 4A, Supplementary Fig. 7).

The initial rate of formation of inversion product was ~40-fold higher than the initial rate of deletion product formation, consistent with predominant 'inversion-ready' alignment of the sites in the synaptic complex at early time points (Fig. 4B). A maximum of 30.8 ± 2.6% inversion product was observed after reaction for 120 min, which was considered to be quite efficient; conversion to inversion product is not expected to exceed 50% because further rounds of recombination can restore the non-recombinant sites. The inversion product could be identified by the altered colour of the gel band (lower green: red ratio; Fig. 4A), indicating higher FRET and confirming that this substrate could be used to follow the mechanism by observing FRET efficiency changes.

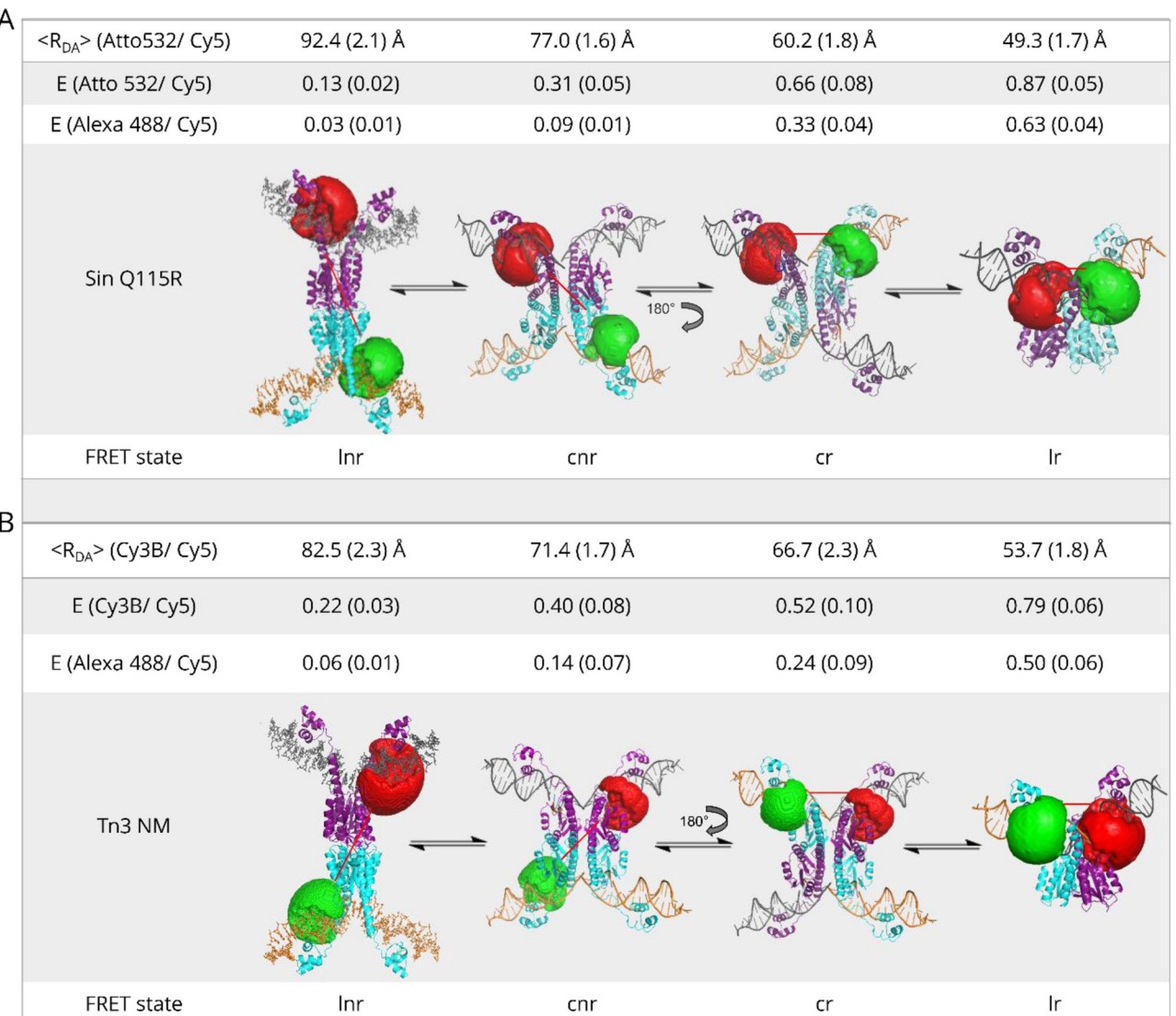

| A | | | | |
|---|---|---|---|---|
| $<R_{DA}>$ (Atto532/ Cy5) | 92.4 (2.1) Å | 77.0 (1.6) Å | 60.2 (1.8) Å | 49.3 (1.7) Å |
| E (Atto 532/ Cy5) | 0.13 (0.02) | 0.31 (0.05) | 0.66 (0.08) | 0.87 (0.05) |
| E (Alexa 488/ Cy5) | 0.03 (0.01) | 0.09 (0.01) | 0.33 (0.04) | 0.63 (0.04) |
| FRET state | lnr | cnr | cr | lr |

| B | | | | |
|---|---|---|---|---|
| $<R_{DA}>$ (Cy3B/ Cy5) | 82.5 (2.3) Å | 71.4 (1.7) Å | 66.7 (2.3) Å | 53.7 (1.8) Å |
| E (Cy3B/ Cy5) | 0.22 (0.03) | 0.40 (0.08) | 0.52 (0.10) | 0.79 (0.06) |
| E (Alexa 488/ Cy5) | 0.06 (0.01) | 0.14 (0.07) | 0.24 (0.09) | 0.50 (0.06) |
| FRET state | lnr | cnr | cr | lr |

**Fig. 3 | AV simulations predicting distances between the fluorophores at different stages of recombination.** The accessible volumes (AV) of the donor fluorophore (green) and the acceptor fluorophore (red) are illustrated as surfaces[48]. The dye pairs used in our experiments are indicated on the Figure. The FRET-averaged donor-acceptor distances, $<R_{DA}>$, were determined using Atto 532/ Cy5 (SUL25 + Sin Q115R) and Cy3B/ Cy5 (UL25 + Tn3 NM) FRET pairs and are shown with standard deviation (SD) in brackets. The expected E values for the Alexa 488/Cy5 FRET pair are also shown. Donor and acceptor positions are illustrated for each conformation expected during recombination (lnr, ligated non-recombinant; cnr, cleaved non-recombinant; cr, cleaved recombinant; lr, ligated recombinant). Note that the lnr structures shown for both Tn3 and Sin are models based on existing crystal structures, as high-resolution structures for these intermediates have not yet been solved. **A** (Sin Q115R): The models are based on the same structures as for Tn3 NM resolvase, but the fluorophores are at their positions in the SUL25 substrate. **B** (Tn3 NM resolvase): The lnr model was created previously by juxtaposition of two γδ resolvase-site I structures (1GDT)[7,11]. The cnr and cr models are based on the crystal structure of a γδ resolvase-site I cleaved synaptic intermediate (1ZR4)[7,11], and the LR model is based on a single 1GDT structural unit. See text for details.

Multiparameter fluorescence detection (MFD) is a single-molecule technique which uses a confocal microscope to record fluorescence as a function of polarization, wavelength, lifetime and macrotime[29,30]. We used MFD to measure FRET within freely diffusing linked-site complexes labelled with Alexa 488/Cy5. No FRET population was observed for SUL25 in the absence of Sin, as expected (Supplementary Fig. 8A). Synapses of SUL25 with Sin were prepared at 20 nM SUL25 DNA concentration and incubated at 37 °C before diluting to picomolar level. To assess whether formation of the synapse without DNA cleavage or recombination reduced the donor-acceptor distance enough to induce an observable FRET signal, we used the catalytically defective Sin mutant Q115R/S9A/R69S. This mutant was previously shown to form synaptic complexes with short site I substrates[31] and formed a synapse with SUL25 on a polyacrylamide gel (Supplementary Fig. 9). However, no FRET was observed (Fig. 4C)

which we attribute to the fluorophores in the uncleaved synapse being too far apart (Fig. 3A). To assay recombination, SUL25/Sin Q115R synapses were again prepared at 20 nM SUL25 DNA concentration and incubated at 37 °C before diluting to picomolar level at 5 min and 120 min time points. A high-FRET population with E = 0.46 (0.01) was observed which increased over time (Supplementary Figs. 8B, C). The high-FRET population increased with $Mg^{2+}$ concentration (compare Supplementary Fig. 8C and Fig. 4D), consistent with faster ligation rate ($Mg^{2+}$ may facilitate the activation of the 3′-OH nucleophile[10,31,32]). We assign this high-FRET population as the ligated recombinant product since its FRET efficiency matches that of recombinant duplex DNA assembled from the corresponding fluorophore-labelled oligonucleotides (Supplementary Fig. 8E).

We conducted equivalent studies using Tn3 NM resolvase and substrate UL25. UL25/Tn3 NM resolvase-catalysed recombination

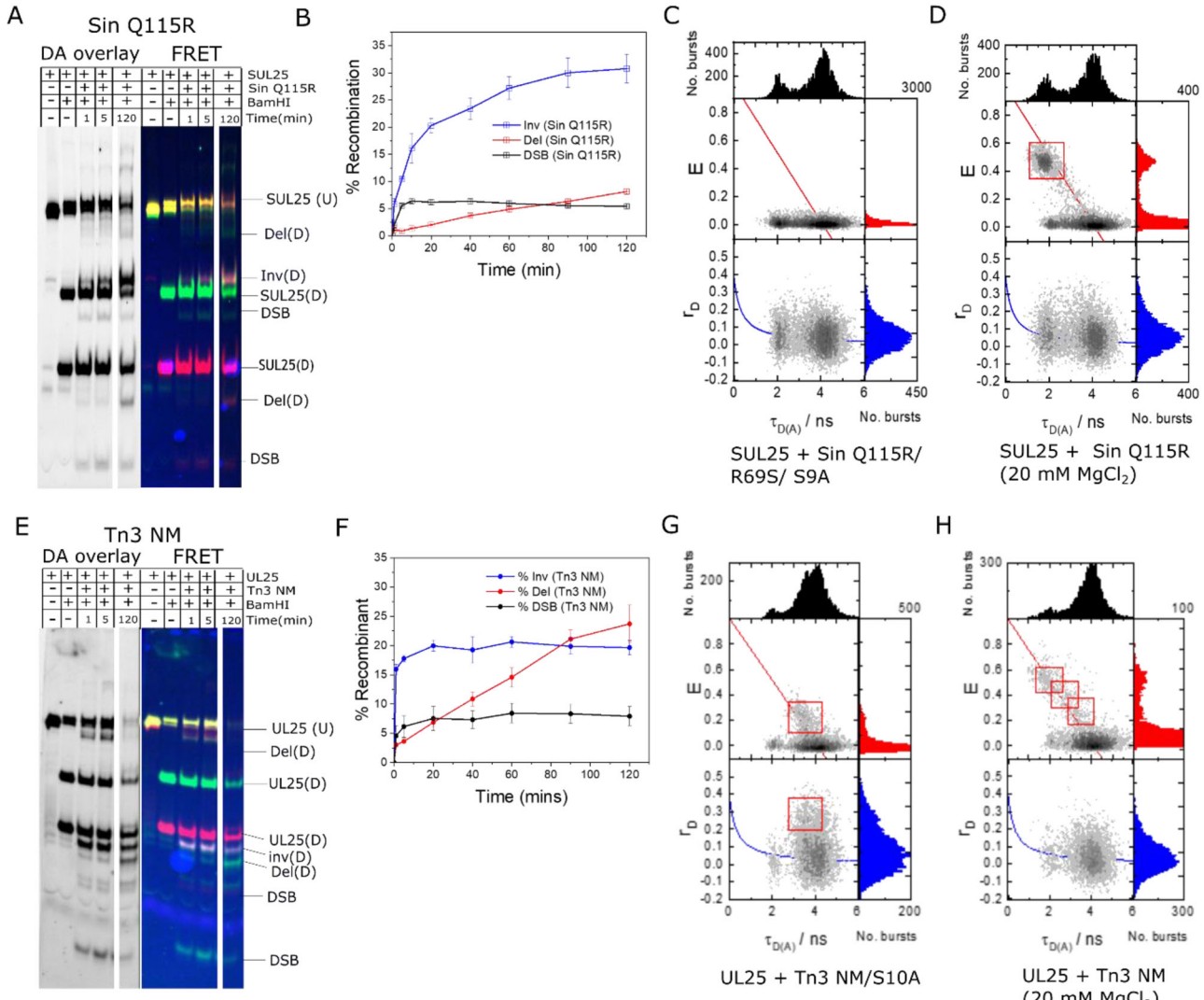

**Fig. 4 | Ensemble and smFRET analysis of reactions with freely diffusing linked site I substrates. A** SUL25 was reacted with Sin Q115R for the indicated times, and the products were digested with BamHI (see Supplementary Information for details). The products were separated by SDS-PAGE. The BamHI digest was incomplete. Bands are labelled as follows: SUL25 (U), undigested substrate (with both green and red fluorophores, giving a yellow band in the overlay); SUL25 (D), digested substrate (with one fluorophore (green or red) in each fragment); Del (D), digested deletion products; Inv (D), digested inversion products (band pink/ white due to high FRET); DSB, double-strand break products (cleaved reaction intermediates; see Fig. 2C). The image on the left of each panel shows combined fluorescence from donor and acceptor scans in greyscale. The 'FRET scan' (coloured image on the right in each panel) was generated as described in 'Methods'. For the full gels see Supplementary Fig. 6. **B** Time course of Sin Q115R/SUL25 recombination showing formation of inversion, deletion and DSB products (%). Error bars represent the standard error of the mean taken from triplicate repeats of recombination assays as in part (**A**). **C** Multiparameter fluorescence detection (MFD) showing a 2D plot of FRET efficiency (**E**) or anisotropy ($r_D$) vs. donor lifetime ($\tau_D$) of freely diffusing SUL25 treated with inactive Sin mutant Q115R/ R69S/ S9A for

10 min. The red overlaid line represents the theoretical FRET relationship $E = 1 − (\tau_{DA} / \tau_D)$ for Alexa 488 with $\tau_D = 4.0$ ns. The blue overlaid line is the Perrin equation, $r_D = r_0/ (1 + \tau_{D(A)} / \rho_D)$, with mean rotational correlation time $\rho_D = 0.35$ ns and fundamental anisotropy $r_0 = 0.375$. Only zero-FRET species were observed. **D** SUL25 was treated with Sin Q115R in buffer containing 20 mM $MgCl_2$ for 120 min. One new FRET population (red box) was clearly observed, with $E = 0.46$ (0.01). **E** UL25 was reacted with Tn3 NM resolvase for 120 min, and digested with BamHI, then analysed by SDS-PAGE. Bands are labelled as in part (**A**). **F** Time course of NM resolvase/UL25 recombination, showing formation of inversion, deletion and DSB products (%). Deletion was the major reaction pathway. Error bars represent the standard error of the mean taken from triplicate repeats of recombination assays as in part (**E**). **G** UL25 was treated with inactive mutant Tn3 NM/S10A for 10 minutes. As well as zero-FRET molecules, a low-FRET population (red box) was observed with $E = 0.24$ (0.01). **H** UL25 was treated with Tn3 NM resolvase for 120 min in the presence of $MgCl_2$. Three non-zero FRET states (red boxes) were observed, with $E = 0.24$ (0.01), 0.52 (0.01), and 0.37 (0.01). See text for further details. Source data are provided as a Source Data file.

(Fig. 4E, F) was initially faster than SUL25/Sin Q115R recombination (Fig. 4A, B). As with Sin, the initial rate of inversion was faster than deletion (-3-fold), again consistent with the $T_{25}$ linker directing synapsis/recombination to proceed with 'inversion-ready' alignment. However, deletion products predominate at later time points, presumably due to repeated rounds of recombination; whereas the inversion product can recombine again (restoring the substrate DNA configuration), the deletion products (two separate DNA molecules;

see Fig. 2C) might dissociate, preventing further recombination and thus accumulating.

The UL25/Tn3 NM resolvase reaction was then studied in solution using MFD. No FRET populations were observed in the absence of resolvase, as expected (Supplementary Fig. 10B), but a low-FRET population with $E = 0.24$ (0.01) was observed following synapsis with the catalytically defective mutant Tn3 NM/S10A[28] (Fig. 4G), in contrast to our results with the catalytically defective Sin mutant Q115R/

**Table 1 | Comparison of experimentally obtained $R_{DA}$ values and those obtained from AV simulations**

| SR | State | D/A | E | $R_0$ | $R_{DA}$ (Å) | E (AV/ γδ) | $<R_{DA}>$(Å) (AV/ γδ) |
|---|---|---|---|---|---|---|---|
| Sin Q115R (37 °C) | lnr | Atto532/ Cy5 | 0.16 (0.04) | 67.4 | 88.9 (4.5) | 0.13 (0.02) | 92.4 (2.1) |
| Sin Q115R (37 °C) | cnr | Atto532/ Cy5 | 0.29 (0.02) | 67.4 | 78.2 (1.3) | 0.31 (0.05) | 77.0 (1.6) |
| Sin Q115R (37 °C) | cr | Atto532/ Cy5 | 0.50 (0.03) | 67.4 | 67.4 (1.3) | 0.66 (0.08) | 60.2 (1.8) |
| Sin Q115R (37 °C) | lr | Atto532/ Cy5 | 0.77 (0.04) | 67.4 | 55.1 (2.1) | 0.87 (0.05) | 49.3 (1.7) |
| Sin Q115R (37 °C, 20 mM Mg) | lnr | Atto532/ Cy5 | 0.13 (0.04) | 67.4 | 92.5 (5.7) | 0.13 (0.03) | 92.4 (2.1) |
| Sin Q115R (37 °C, 20 mM Mg) | cnr | Atto532/ Cy5 | 0.29 (0.03) | 67.4 | 78.2 (2.5) | 0.31 (0.05) | 77.0 (1.6) |
| Sin Q115R (37 °C, 20 mM Mg) | cr | Atto532/ Cy5 | 0.53 (0.03) | 67.4 | 66.1 (1.3) | 0.66 (0.08) | 60.2 (1.8) |
| Sin Q115R (37 °C, 20 mM Mg) | lr | Atto532/ Cy5 | 0.79 (0.04) | 67.4 | 54.6 (2.2) | 0.87 (0.05) | 49.3 (1.7) |
| Sin Q115R (21 °C) | lnr | Atto532/ Cy5 | 0.17 (0.02) | 67.4 | 87.8 (4.2) | 0.13 (0.03) | 92.4 (2.1) |
| Sin Q115R (21 °C) | cnr | Atto532/ Cy5 | 0.34 (0.02) | 67.4 | 75.3 (2.2) | 0.31 (0.05) | 77.0 (1.6) |
| Sin Q115R (21 °C) | cr | Atto532/ Cy5 | 0.54 (0.05) | 67.4 | 65.6 (4.4) | 0.66 (0.08) | 60.2 (1.8) |
| Sin Q115R (21 °C) | lr | Atto532/ Cy5 | 0.78 (0.03) | 67.4 | 54.6 (3.2) | 0.87 (0.05) | 49.3 (1.7) |
| Tn3 NM | lnr | Cy3B/ Cy5 | 0.12 (0.03) | 67.0 | 93.4 (4.5) | 0.22 (0.03) | 82.5 (2.3) |
| Tn3 NM | cnr | Cy3B/ Cy5 | 0.31 (0.03) | 67.0 | 76.6 (1.8) | 0.40 (0.08) | 71.4 (1.7) |
| Tn3 NM | cr | Cy3B/ Cy5 | 0.54 (0.03) | 67.0 | 65.2 (1.3) | 0.52 (0.10) | 66.7 (2.3) |
| Tn3 NM | lr | Cy3B/ Cy5 | 0.73 (0.02) | 67.0 | 56.7 (1.0) | 0.79 (0.06) | 53.7 (1.8) |
| Sin Q115R (MFD) | lnr | Alexa488/ Cy5 | – | 51.4 | – | 0.03 (0.01) | 92.4 (2.1) |
| Sin Q115R (MFD) | cnr | Alexa488/ Cy5 | – | 51.4 | – | 0.09 (0.01) | 75.8 (1.6) |
| Sin Q115R (MFD) | cr | Alexa488/ Cy5 | – | 51.4 | – | 0.33 (0.04) | 57.8 (1.6) |
| Sin Q115R (MFD) | lr | Alexa488/ Cy5 | 0.50 (0.02) | 51.4 | 52.8 (0.7) | 0.63 (0.04) | 47.2 (1.8) |
| Sin Q115R (20 mM Mg) (MFD) | lr | Alexa488/ Cy5 | 0.50 (0.02) | 51.4 | 52.8 (0.7) | 0.63 (0.04) | 47.2 (1.8) |
| Tn3 NM-S10A (inversion) (MFD) | lnr | Alexa488/ Cy5 | 0.23 (0.01) | 51.4 | 62.9 (0.6) | 0.06 (0.01) | 80.7 (2.3) |
| Tn3 NM-S10A (deletion) (MFD) | lnr | Alexa488/ Cy5 | 0.23 (0.01) | 51.4 | 62.9 (0.6) | 0.10 (0.01) | 74.1 (2.3) |
| Tn3 NM (MFD) | cnr | Alexa488/ Cy5 | 0.24 (0.01) | 51.4 | 62.3 (0.6) | 0.14 (0.02) | 66.4 (1.6) |
| Tn3 NM (MFD) | cr | Alexa488/ Cy5 | 0.37 (0.01) | 51.4 | 56.2 (0.4) | 0.24 (0.05) | 62.1 (2.2) |
| Tn3 NM (MFD) | lnr | Alexa488/ Cy5 | 0.5 (0.01) | 51.4 | 51.4 (0.3) | 0.50 (0.05) | 51.9 (1.7) |

Donor-acceptor distances ($R_{DA}$) for SUL25/Sin Q115R complexes in the presence and absence of $Mg^{2+}$, and UL25/Tn3 NM complexes with no $Mg^{2+}$, were calculated using the Förster equation using values of E obtained from HaMMy (TIRF) or Gaussian fitting (MFD). The Forster radii used were, for Atto 532/ Cy5, $R_0$ = 67.4 Å[50,51]; for Alexa 488/ Cy5, $R_0$ = 51.4 Å[52,53]; and for Cy3B/ Cy5, $R_0$ = 67 Å[54]. $<R_{DA}>E$ is the FRET-averaged distance between the dyes, resulting from integrating overall possible FRET efficiencies determined through AV simulation of various fluorophore positions modelled on crystal structures of γδ-resolvase[7,9]. Standard deviations (given in parentheses) in experimental values of E were determined through clustering analysis of the transition density plots in MASH-FRET[34] or were determined by AV simulation.

S9A/R69S where no similar low-FRET population was detectable. We suggest this (small) population might correspond to a 'pre-deletion' synaptic complex (see Fig. 2C), as our AV simulations (see above) predict that the fluorophores are closer ($R_{DA}$ = 76.5 (2.3) Å, $E_{Alexa\ 488/Cy5}$ = 0.10 (0.04)) than in the pre-inversion synapse ($R_{DA}$ = 82.5 (2.3) Å, $E_{Alexa\ 488/Cy5}$ = 0.06 (0.03); Supplementary Fig. 11). Alternatively, the population might be a pre-inversion synapse, but with a shorter DA distance than in the Sin synapse due to the conformation of the Tn3 resolvase synaptic tetramer and smaller distance between fluorophores (Fig. 3B)[21]. We observed a substantial increase in donor anisotropy for the UL25/Tn3 NM/S10A low-FRET population (Fig. 4G), suggesting that the fluorophore rotational mobility becomes restricted upon resolvase binding. Analysis of the MFD data from a UL25/Tn3 NM resolvase recombination assay revealed two additional FRET populations (Fig. 4H); the high-FRET population (E = 0.50 (0.01)) is consistent with the AV simulation of the ligated recombinant (lr) complex (Supplementary Fig. 10D), and the intermediate-FRET population (E = 0.37 (0.01)) correlates best with the AV simulation for the cleaved intermediate in the recombinant configuration (cr) (Table 1).

As an alternative approach to studying subunit rotation, we designed a FRET assay in which one of the fluorophores was attached to Tn3 NM resolvase. A Cy5 was attached at position 185, the C-terminal residue (see Methods section for details), and a FRET donor (Alexa 488) was attached to the DNA substrate. A Cy5-labelled resolvase subunit could be incorporated at one of four positions, creating various possible FRET outcomes (Supplementary Fig. 12A), which we

observed in the MFD data (Fig. S12C, D). Since synapses that contain more than one labelled subunit would result in additional FRET states and a much more complicated kinetic analysis (see below), we were unable to investigate further and the experiments with labelled resolvase serve primarily as further evidence for the controlled formation of a substrate-bound recombinase tetramer.

**Recombination of surface-immobilized synaptic complexes**
While the confocal (MFD) approach presented above shows the distribution of FRET states in solution, it does not report on the dynamics of recombination. We employed Total Internal Reflection Fluorescence (TIRF) microscopy to study the dynamics of immobilized linked site I substrates undergoing recombination by resolvase mutants (Fig. 5). The UL25 substrate for Tn3 NM resolvase had a Cy3B FRET donor and a Cy5 acceptor, whereas the SUL25 substrate for Sin Q115R had an Atto 532 donor and a Cy5 acceptor. Note that these FRET pairs differ from those used in the MFD experiments, resulting in small differences in the $R_{DA}$ values (Fig. 3, Table 1). The biotin label for surface attachment of the substrates was positioned at the 5' end of the strand bearing the Cy5 acceptor fluorophore (Supplementary Fig. 3), to ensure that only properly assembled substrates could be imaged by donor excitation. When the substrate SUL25 was attached to the surface in the absence of Sin, a constant FRET signal close to zero was observed, as expected (Supplementary Fig. 13A). A construct corresponding to the SUL25 inversion product (HFSUL25) also gave a constant FRET signal in the absence of Sin Q115R, but in this case the FRET was high, as expected (E = 0.72) (Supplementary Fig. 13B).

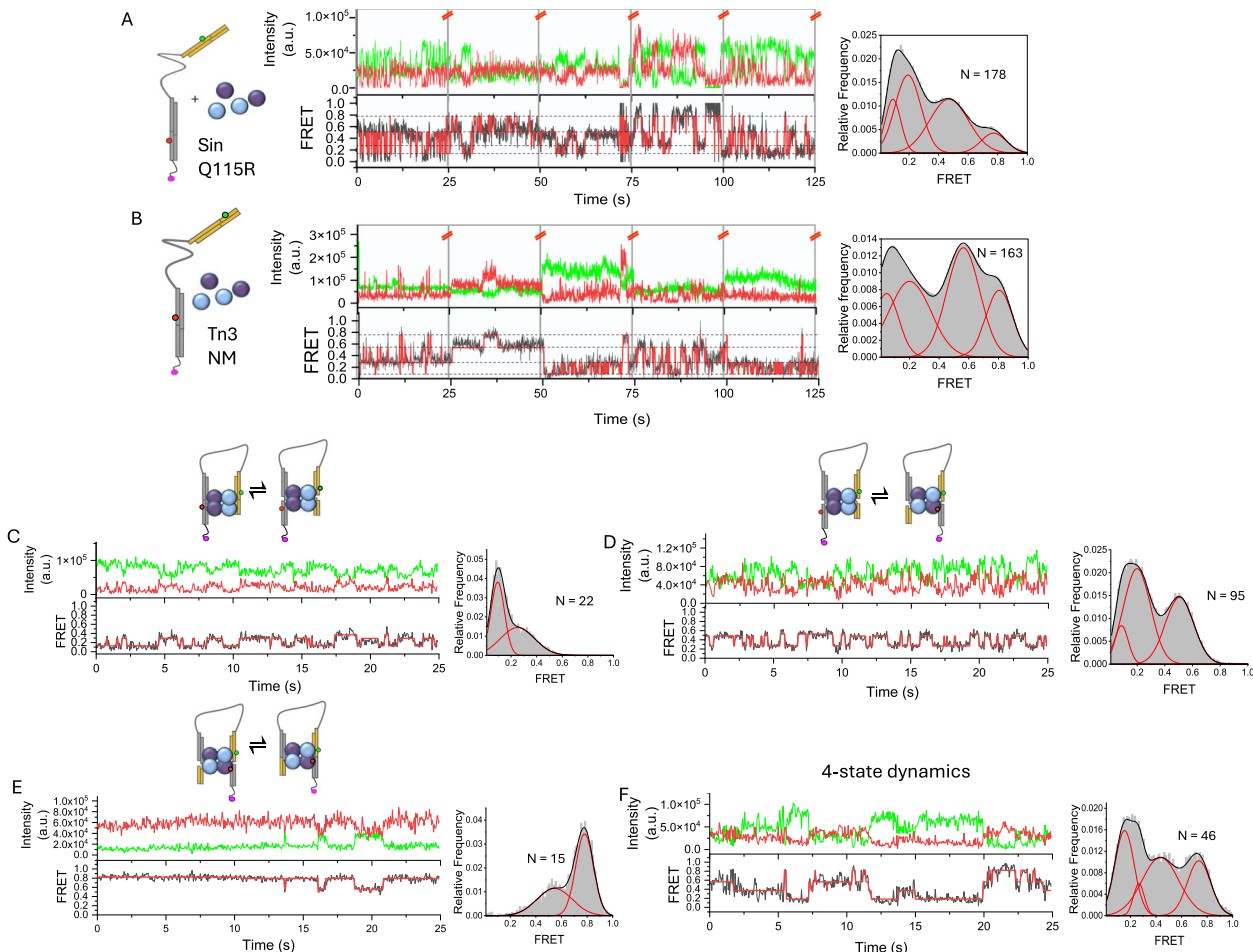

**Fig. 5 | smFRET trajectories of immobilized substrates.** Donor (green) and acceptor (red) intensity traces are shown in the top panels, and FRET traces (black) and fit (red) after Hidden Markov Modelling (HMM) are shown in the bottom panels. **A** Stitched (after every 25 s, indicated by dividers and red lines) time trajectories of surface-immobilized SUL25 incubated with Sin Q115R at 37 °C. The red overlaid line shows the result from HMM analysis fit to 4 states. The FRET histograms of all filtered traces (on the right) were fit to Gaussian curves giving E values

of 0.09, 0.19, 0.47, and 0.77. These values are in good agreement with predicted E values for Sin Q115R 'inversion' shown in Fig. 3A. **B** Analysis as in (**A**) is shown for UL25 recombination with Tn3 NM. The Gaussian fit gave E = 0.05, 0.20, 0.56, 0.81, which also had good correlation with predicted E values shown in Fig. 3B. **C–F** TIRF time trajectories of surface-immobilized SUL25 in the presence of Sin Q115R at 37 °C and their respective FRET histograms. (**C**) lnr ↔ cnr transitions, (**D**) cnr ↔ cr, (**E**) cr ↔ lr, (**F**) one molecule showing transitions between all states.

Experiments with SUL25 and Sin Q115R resolvase were initially conducted at 37 °C in the absence of MgCl₂. Synapses were prepared at nanomolar concentration (see Methods section) prior to dilution and immobilization. No MgCl₂ was added to the buffer, to maximize the stability of the complexes and encourage multiple rounds of rotation[28]. All dynamic trajectories were analysed with the software packages HaMMy[33] and MASH-FRET[34], using a method similar to that described previously by Bianco et al.[35]. Unstitched time trajectories were used to produce transition density plots (TDPs), and these were analysed using a clustering algorithm, which fits the TDP to a mixture of K (K − 1) isotropic 2D Gaussians to obtain the optimal number of FRET states ($K_{opt}$). We found that $K_{opt}$ = 4 was consistent with both the low value of the Bayesian information criterion (BIC) (Fig. 6B) and with our physical model (Fig. 3). A $K_{opt}$ of 4 was also consistent with the FRET data for Sin recombination at room temperature (21 ± 1 °C) (Supplementary Fig. 14) and at 37 °C in the presence of MgCl₂ (Supplementary Fig. 15). FRET E values were derived from the Gaussian means and the associated errors (in parentheses) from the average Gaussian sample standard deviations. The TDP showed the presence of multiple interconverting FRET states (Fig. 6C), which we assign to the predicted recombination (inversion) intermediates (Fig. 3) as follows: $E_{lnr}$ = 0.16 (0.04), $E_{cnr}$ = 0.29 (0.02), $E_{cr}$ = 0.50 (0.03) and $E_{lr}$ = 0.77 (0.04). A consistently

high number of transitions were observed between cnr and cr states for Sin recombination at 37 °C with and without Mg²⁺ and at room temperature (Fig. 6C, Supplementary Figs. 14 and 15). These transitions were reversible and fast (Fig. 5D; see Supplementary Fig. 16 for further examples). We also found a relatively higher number of transitions between 'early-stage' conformations (lnr ↔ cnr) at room temperature than at 37 °C, in agreement with ensemble data (Supplementary Figs. 2, 14D).

Our assignment of the FRET states in the SUL25 + Sin Q115R experiments is in broad agreement with the predicted E values from the AV simulations (Fig. 3) and the observed FRET E value of the naked SUL25 recombinant (Supplementary Fig. 13B). We discuss possible reasons for discrepancies between the observed and modelled data below (see Discussion).

TIRF experiments were also conducted using UL25−Tn3 NM resolvase synapses at room temperature, in the absence of Mg²⁺ ions. As with Sin Q115R, we assigned the lowest E state to the lnr inversion intermediate and the slightly higher E state to the cnr intermediate in our kinetic model, as both had good agreement with AV simulations. A large number of transitions occurred between the cnr state ($R_{DA}$ = 76.6 (1.8) Å) and the cr state ($R_{DA}$ = 65.2 (1.3) Å) (Fig. 5B, Supplementary Fig. 17D), similarly to the SUL25/Sin Q115R data. Our observations are

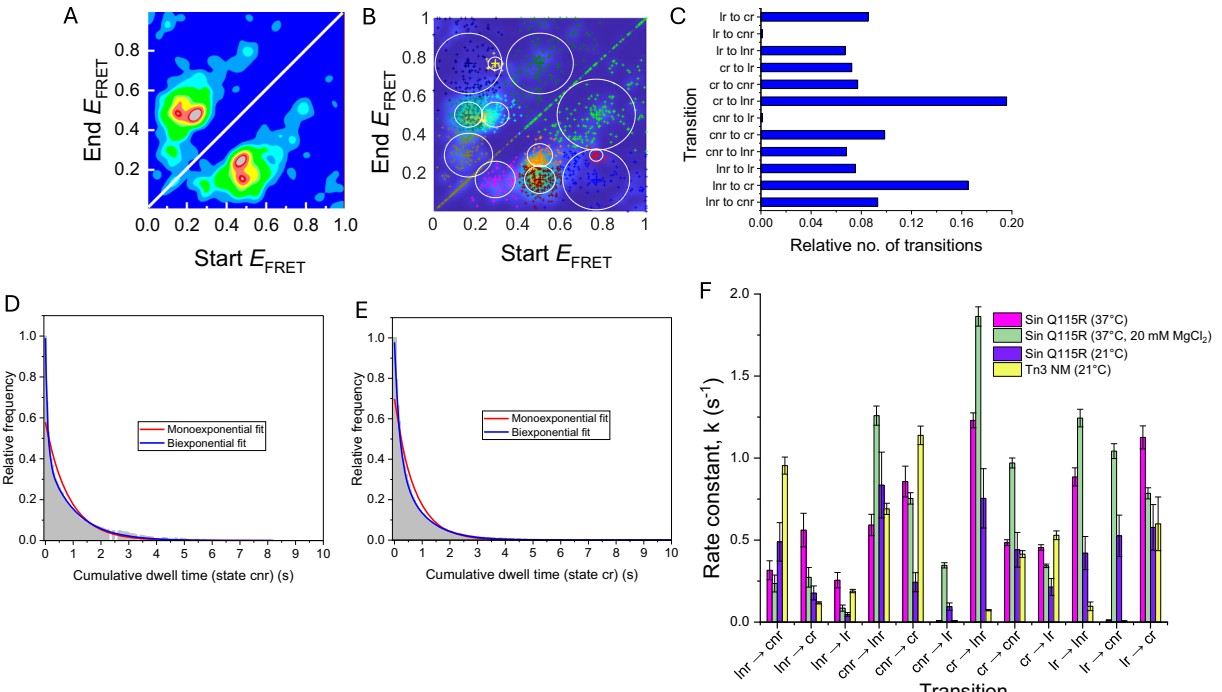

**Fig. 6 | Transition analysis of immobilized SUL25 substrates in the presence of Sin Q115R. A** A Gaussian-convoluted transition density plot (TDP) from unstitched dynamic time trajectories of SUL25 in the presence of Sin Q115R. **B** A clustering algorithm is applied to determine the overall state configuration. The most sufficient cluster model is determined by the BIC, which here has the lowest value when the number of states (J) = 4[34], with $E_1$ = 0.16 (0.04), $E_2$ = 0.29 (0.02), $E_3$ = 0.50 (0.03) and $E_4$ = 0.77 (0.04). **C** Results from 4-state model inference. The relative number of transitions for all possible states is shown (blue bars). **D, E** Exponential (red) and biexponential (blue) decay functions fit to cumulative dwell time histograms for both cnr (**D**) and cr (**E**) states. **F** Comparison between rate constants of each possible transition observed for Sin Q115R-catalysed recombination in the absence (at 21 and 37 °C) and presence of $Mg^{2+}$ (37 °C) and for Tn3 NM catalysed recombination (no $Mg^{2+}$ at 21 °C). The error bars represent the weighted average of the biexponential fitting parameters determined using the bootstrapping feature in BOBA-FRET (3× standard deviation for 100 bootstrapped samples); see 'Materials and Methods' for further details.

consistent with 180° rotations from cnr to cr, in an open cleaved state resembling the 1ZR4 crystallographic structure of the γδ resolvase intermediate. These transitions occurred randomly and processively (Supplementary Fig. 18B). The most frequent transitions were between the cr state ($R_{DA}$ = 65.2 (1.3) Å) and lr state ($R_{DA}$ = 56.7 (1.0) Å) (Supplementary Fig. 17D), consistent with a greater number of molecules being observed at a later stage in the recombination reaction, as expected from the results of our ensemble experiments, where equilibrium was reached after only 5–10 min (Fig. 4F).

For the deletion pathway (expected to occur in a small fraction of the observed molecules; see Fig. 4) the predicted FRET changes would be from E - 0.38 (uncleaved synapse; Supplementary Fig. 11) to a higher E value upon cleavage, but no change in FRET would be observed during rotations. However, we cannot unambiguously assign any FRET signals to deletion intermediates.

Rate constants for state-to-state transitions were determined by averaging the k values from biexponential fits to cumulative dwell time histograms generated from the TIRF data (Fig. 6D, E, Supplementary Fig. 19). The rate constants for the rotation steps were computed for Sin recombination at 37 °C as 0.86 (0.09) s⁻¹ (state cnr to state cr) and 0.48 (0.02) s⁻¹ (state cr to cnr) and Tn3 NM recombination as 1.14 (0.06) s⁻¹ (state cnr to state cr) and 0.41 (0.02) s⁻¹ (state cr to state cnr). For both resolvases, single complexes show multiple transitions over the ~25 s period of observation, indicating the stability of the cleaved site I-resolvase synapse. We observed faster cnr to cr rotation rates for Sin recombination upon increasing temperature from room temperature to 37 °C, and for Tn3 NM compared with Sin Q115R, agreeing with our observations from ensemble studies (Fig. 6F).

We also observed transitions in both systems that apparently skipped intermediate states, such as transitions from cnr ↔ lr and lnr

↔ lr (Figs. 6C, Supplementary Figs. 14D, 15D, 17D). Skipped states can result from fast intermediate transitions with dwell times approaching the time of each 50 ms TIRF movie frame, causing them to be unresolved in HMM analysis[36] (for example, an apparent direct transition from a hypothetical state 1 to 3 could be a composite of transitions from state 1 to 2 and then 2 to 3). Interestingly, we observed better fits of the majority of our dwell-time distributions using a biexponential decay function (Supplementary Fig. 19), resulting in one relatively fast decay component. For example, the cnr state (Sin Q115R, 20 mM $MgCl_2$) has $\tau_1$ = 80 ms (Supplementary table 3) approaching the frame rate used (50 ms), supporting the idea that skipped states are the result of unresolved, fast intermediate transitions. All rate constants derived from biexponential decay fittings are shown in Supplementary Table 3. Notably, the biexponential fits imply kinetic heterogeneity in the data for both resolvases. An intriguing possibility is that this kinetic heterogeneity reflects a dependence on the direction of subunit rotation (see Discussion).

## Discussion

The subunit rotation mechanism proposed for serine recombinases is unique in enzymology in that one-half of the entire protein-DNA recombination intermediate rotates through 180° (or multiples of 180°) relative to the other half, with no obvious structural constraint on catastrophic dissociation of the two halves (Fig. 1). The model has therefore been controversial since it was first proposed in the 1980s (ref. 8 and references cited therein), based on observations of changes to plasmid DNA topology brought about by recombination. Early crystal structures, including that of a γδ resolvase dimer bound to site I of *res*[7], showed an interdigitated dimer interface that appeared to be incompatible with a rotation mechanism. However, subsequent

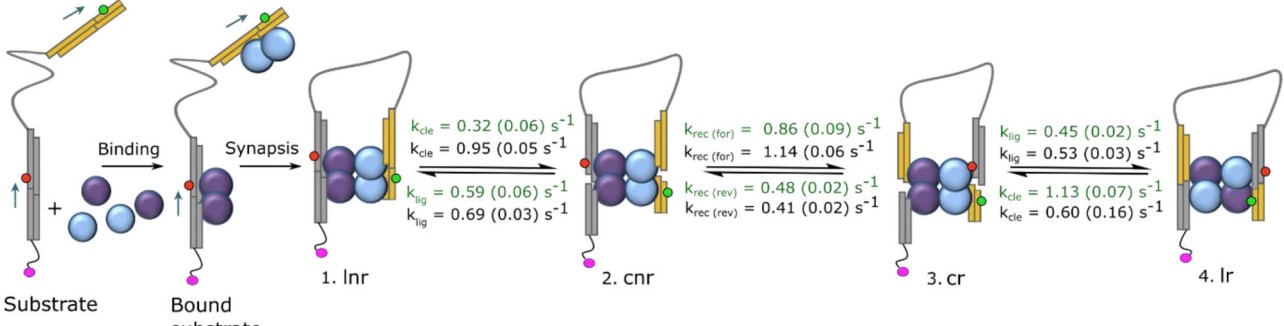

**Fig. 7 | The kinetic pathway model for resolvase-mediated inversion on linked site I substrates.** The substrate and resolvase subunits are cartooned as in Fig. 1. The free substrate and bound but non-synapsed substrate produce no FRET signal and thus could not be observed in the TIRF analysis. The model inferred from our single-molecule analysis includes transitions between ligated non-recombinant (lnr) synapse, cleaved non-recombinant (cnr), cleaved recombinant (cr) and ligated recombinant (lr). Rates for Sin Q115R recombination (green) and Tn3 NM recombination (black) of each transition are depicted in the scheme, including 'forward' and 'reverse' rotation rates ($k_{rec}$, $k_{non-rec}$, respectively), and the rates of cleavage and ligation ($k_{cle}$ and $k_{lig}$). Rate values were extracted from biexponential TIRF analysis of recombination (see Supplementary Figs. 17A–F for Tn3 analysis).

crystallography revealed the structures of intermediates comprising a γδ resolvase tetramer bound to two DNA recombination site segments, with each resolvase subunit covalently attached to a DNA strand 5′ end by its active site serine residue, creating a double-strand break at the centre of each site exactly as predicted by earlier biochemical studies[9,20] (Fig. 1A). Remarkably, the dimer interfaces have been transformed in these structures, and the two 'halves' of the complex that are predicted to rotate relative to each other are held together by a very flat hydrophobic interface between the 'rotating dimers' of resolvase subunits. Molecular modelling showed that rotation whilst conserving this interface should be feasible without any substantial energy barriers. Subsequent crystal structures of other serine recombinase catalytic domain tetramers have revealed similar interfaces between the rotating dimers but in multiple rotational states, consistent with facile continuous rotation, and further support for the model has come from biochemical analyses, including single-molecule magnetic beads experiments (reviewed in refs. 2–4). However, a full understanding of the mechanism requires details of the kinetics of individual steps, especially the crucial rotation step per se that follows double-strand cleavage of each recombination site by the recombinase catalytic tetramer. We therefore undertook the analysis reported here, where we hoped to observe individual steps in real-time by correlating them to changes in FRET efficiency.

For our FRET studies, we used variants of two well-characterized serine recombinases: Tn3 resolvase, which is almost identical to γδ resolvase, the protein used in the first crystallographic studies (see above), and Sin, for which crystallography reveals striking structural similarity with Tn3/γδ resolvase despite primary sequence divergence. In both cases, we used 'activated' mutants of the wild-type recombinases that were selected for loss of dependence on accessory factors and that promote efficient recombination of oligonucleotide substrates[3,31,37].

Our smFRET data are consistent with a 4-state model describing transformations of the initial non-recombinant synaptic complex as it undergoes cleavage, rotation and ligation (corresponding to the cartoons in Fig. 1A). For both Tn3 NM and Sin systems we assign the four FRET states in order from low to high FRET efficiency to the intermediates labelled lnr, cnr, cr and lr in Fig. 3. Our assignment is generally well supported by the fluorophore AV simulations that predict $R_{DA}$ and FRET E values from available structural data (Fig. 3). These predictions are likely to be most reliable for the Tn3 resolvase FRET data, as our models of the intermediates are based on crystallographic structures of γδ resolvase, a very close relative of Tn3 resolvase, in complexes with site I DNA. No equivalent structures of Sin resolvase with its site I DNA are available, so our modelling of the Sin

intermediates is based on the same (γδ resolvase) structures. The assignments of the cnr and cr states are also strongly supported by the observation of frequent reversible cnr ↔ cr transitions, as expected for the 'rotating' intermediates, and assignments of the lr states correspond with the observed E values for synthetic recombinant substrates (Supplementary Fig. 13B).

Our data, summarized in the kinetic scheme shown in Fig. 7, indicate that subunit rotation can occur on a sub-second time scale without intermediate pauses. The complexes remain over longer periods in 'rest states' which we interpret to be double-strand cleaved intermediates in either 'non-recombinant' or 'recombinant' rotational configurations. We suggest that the stability of these intermediates might be due to transitory base-pair formation at the centres of the cleaved recombination sites (see below). The rates of transitions between the states for the Sin and Tn3 systems were quite similar (Supplementary table 3). The rates of rotation (0.4–1.1 s$^{-1}$) are >100-fold slower than those reported previously for small serine invertases and serine integrases in magnetic tweezer experiments[26], but the differences might be explained by the differing methodologies. In the magnetic tweezers experimental systems, loss of supercoiling provides free energy for multiple rounds of unidirectional rotation, and DNA torsional stress might enhance the rates of catalysis. Our oligonucleotide linked-site substrates are designed to favour an inversion pathway (Fig. 2) for which the recombinant and non-recombinant configurations are isoenergetic, so there is no thermodynamic 'driving force' for rotation; it is expected to occur by random thermal fluctuations. Additionally, the different proteins used in these studies might have different reaction kinetics.

As noted above, individual molecules show frequent transitions between the cnr and cr states over the ~25 s observation period (Fig. 5D, Supplementary Figs. 16, 18B), consistent with multiple rounds of either left-handed or right-handed 180° rotation. The data would also be consistent with larger rotations of 360° or more; any rapid rotation (with intermediate state dwell times shorter than our 50 ms frame rate) of an odd multiple of 180° would be indistinguishable from a single 180° rotation, and an even multiple of 180° rotations would be FRET-silent (see Results section).

Unexpectedly, we observe sharp 'subunit rotation' transitions (between FRET states cnr and cr), with no evidence within the resolution of our experiments for molecules in intermediate rotational states. We propose that cnr and cr represent fully cleaved intermediates that are rotationally stalled in positions ready for re-ligation of the DNA ends in non-recombinant or recombinant configurations respectively. Li et al. (2005) carried out modelling studies based on their crystallographic structures of γδ resolvase intermediates that

predicted a very flat energy landscape as rotation proceeds[9,20]; also, crystal structures of Sin and Gin tetramers in the absence of DNA in varying rotational states have been reported[21–23], and crosslinking experiments with Hin intermediates also support multiple rotational states[32,38,39]. However, lower-energy states at positions where ligation of the DNA ends can take place would be consistent with other mechanistic analysis. The most obvious structural feature that could hold the intermediates in these positions would be basepairing between the 2-nt single-stranded tails on each cleaved DNA half-site. However, such basepairing was not observed in the γδ resolvase intermediate crystal structures, where the ends that are to ligate are quite far apart, and further conformational changes would be necessary to bring the ends into ligatable positions. It was proposed[13] that ionic interactions between two 'patches' of charged amino acids (D95, D96 and R121, R125) seen in these crystal structures might be important for docking in the pre-ligation positions.

A role for basepairing prior to re-ligation is consistent with a substantial body of data on reactions of substrates with 'mismatched' recombination sites; that is, the central 2-bp 'overlap sequence' of one site is different from that of the other. These substrates do not make more than trace amounts of recombinants, but instead do 'double rounds' of strand exchange equivalent to 360° rotations[4,14] which do not change the DNA sequence, but leave a signature as topological changes of the supercoiled circular DNA substrate molecules. Those data imply a pre-ligation base-paired intermediate[40] that could correspond to our FRET states cnr and cr. Single-molecule TIRF experiments with mismatched substrates might clarify the status of the intermediates, though it should be noted that rapid 360° rotation events would be 'FRET-silent'.

It is also significant that we see many consecutive rotational transitions (cnr to cr and vice versa) in individual complexes without intervening conversion to the low- or high-FRET states corresponding to ligated intermediates. This provides further support for subunit rotation, the only mechanism that can bring about multiple rounds of strand exchange without a 'reset' step which would typically involve re-ligation of the DNA ends[40,41].

Reactions of wild-type Sin and Tn3 resolvase with supercoiled plasmid substrates in vitro proceed with unobservable or only trace amounts of cleaved intermediates, implying that following cleavage of both sites, rotation and re-ligation of the DNA ends is very fast (see for example refs. 8,37). These wild-type systems contain regulatory elements (accessory DNA sequences and protein subunits) which favour termination of the reaction after one round of strand exchange, probably by destabilizing the ligated recombinant synapse[3,42]. Here, we used activated mutants of Sin and Tn3 resolvase to allow for the use of non-supercoiled, oligonucleotide substrates in our single-molecule FRET experiments. These mutants produce much higher levels of cleaved intermediates in reactions of supercoiled or non-supercoiled substrates in vitro. Activating mutations similar to those of the Tn3 NM resolvase variant used here are also present in the γδ resolvase cleaved intermediate crystal structures[9,20]. The mutations are thought to stabilize the site I synaptic intermediates (which might be formed only transiently in the wild-type systems), and thereby might also favour persistence of the cleaved states, as is observed[28,31].

In the natural Tn3 or Sin recombination systems, unidirectional ('right-handed') rotation is favoured by a concomitant energetically favourable reduction in negative supercoiling of the substrate DNA[8,43]. However, the subunit rotation model implies that rotation could be in either sense, and indeed products of wild-type Tn3 resolvase consistent with both right- and left-handed rotation were observed in non-supercoiled substrates[8]. The oligonucleotide DNA substrates used for this work are not supercoiled, so there is no thermodynamic driving force for a specific sense of rotation; random fluctuations should result in either right-handed or left-handed rotation. Also, repeated rounds of recombination between substrate and product DNA configurations

are expected, as is observed in our FRET experiments. Although the direction of motion cannot be distinguished based on the steady-state FRET populations, because the start and end states are identical, this question might be addressed by kinetic analysis. Our smFRET data provide evidence of kinetic heterogeneity at each step of the mechanism, as illustrated in Fig. 6D, E. To examine this further, we modelled 360° subunit rotation in PyMOL, by rotating one-half of the cleaved γδ resolvase tetramer structure relative to the other half in small steps. We performed AV simulations using the FRET positions in UL25 and plotted the predicted DA distances (and resulting E values) at each stage of rotation (Supplementary Fig. 20). Note that at 0° the FRET distance is the same as that of the cnr state for UL25 (Fig. 3). We found that the changes in DA distance predicted to occur during 0°–180° rotation were not exactly reversed during 180°–360° rotation. Thus, right-handed and left-handed subunit rotation events are predicted to give different FRET changes as rotation proceeds. It is tempting to speculate that the structural heterogeneity implied by the biexponential rates (Supplementary Fig. 19) is due to this asymmetry of the synapse (Supplementary Fig. 20), with some synapses preferentially rotating in one or the other direction. This might be due to subtle alterations in the conformations of otherwise identical synapses (e.g. via differences in the ionic environment of the protein or DNA due to metal ion coordination). We may be able to test this in the future by designing synapses that are biased to move in a specific direction.

Our experiments here used two members of the 'small serine recombinase' family (Sin and Tn3 resolvase), but it is hypothesised that all other serine recombinases, including the 'large' bacteriophage serine integrases and serine transposases, promote DNA strand exchange by the same subunit rotation mechanism. The catalytic domain that forms the synaptic tetramer and thus the rotating interface is highly conserved in size and sequence throughout the many thousands of serine recombinases identifiable in databases, suggesting similar functions, and reactions implying subunit rotation have been studied for members of the serine integrase group[16,25]. A crucial feature of natural serine recombinase systems that remains incompletely understood is the mechanism of strand exchange regulation. Natural systems (including the wild-type Tn3 resolvase and Sin systems) typically terminate strand exchange after only one round of 180° rotation, giving the biologically relevant recombination products, even though the mechanism is by its nature repeatable indefinitely, as we observe in our FRET experiments and as has been deduced from earlier experimental studies (see above). Selectivity for a single 180° rotation is intimately connected with the requirement of natural systems for formation of elaborate synaptic complexes involving accessory factors (extra DNA sequences and protein subunits) before the catalytic steps of DNA cleavage and ligation are licensed[2–4,42]. Although regulatory interactions of the catalytic tetramers with accessory factors have been identified and characterized, it is still unclear how these 'signals' are transmitted to control the DNA cleavage, subunit rotation, and DNA ligation steps.

## Methods

### DNA

**Substrate design.** A full list of all oligonucleotides used in this work is shown in Supplementary Table 1. Our understanding of the mechanism (see Fig. 1B, C) indicated that the fluorophores would be best positioned near the centre of site I, one on either side of the site of cleavage and exchange (Fig. 2A, B). The crystal structures of resolvase-DNA complexes show that resolvase does not contact the major groove at several base pairs on either side of the centre of site I, and these are therefore good potential attachment positions for fluorophores attached to bases on their major groove edges. In preliminary ensemble experiments we analysed the efficiency of recombination by Tn3 resolvase (NM activated mutant;[28]) with site I substrates containing fluorophores attached at these positions. The 50-bp

oligonucleotides, each with donor (Alexa 488) and acceptor (Cy5) fluorophores attached to the C-5 methyl of a T base ('Ser1' substrates; Supplementary Table 1) were treated with resolvase in the presence of an excess of an 80-bp non-fluorescent site I oligonucleotide ('Ser 2', Supplementary Table 1). Recombination generates a 65-bp recombinant product that can be distinguished from the substrate by size on a polyacrylamide gel (see Supplementary Fig. 1 for an example). Using this assay we observed efficient recombination of substrates with fluorophore modifications at positions −6B, −5T, +5T and +6T (Supplementary Table 2). Positions −6B and +5T were chosen for fluorophore pair attachment in our FRET experiments (Fig. 2B).

The structures of Sin and Tn3 resolvase bound to their respective site Is are predicted to be very similar[7,9,21,31,37], so we expected that fluorophores attached to Sin site I at analogous positions to those used for Tn3 site I would not interfere substantially with recombination by the activated Sin mutant Q115R[21,31]. However, we could not use exactly the same positions because of differences between the Tn3 site I and Sin site I sequences. Our chosen attachment positions in Sin site I (substrates 'Sinser1' and 'Sinser2') at −6T and +7B permitted efficient recombination (Fig. 2A; Supplementary Fig. 2; Supplementary Table 1).

In preliminary experiments to determine the optimal length of the poly-T linker in our linked-site substrates, we used substrates containing only one fluorescein label at the Y-arm 5'-end (Fig. 2C), and linkers of varying length ($T_{25}$, $T_{40}$ and $T_{50}$). The $T_{25}$ linker gave the highest level of inversion product (Supplementary Fig. 4, Supplementary table 4), so we used a $T_{25}$ linker in subsequent experiments. Restriction enzyme sites were also included in the substrate design to facilitate analysis of reaction products. See Supplementary Table 1 for the sequences of the oligonucleotides used to create the substrates. The linked-site Sin Q115R substrate SUL25 (Fig. 2C) has two copies of Sin site I, each fluorophore-modified (Fig. 2A; Supplementary Fig. 3B); the Tn3 NM resolvase substrate UL25 is similar except that it has two copies of Tn3 site I, and the fluorophore positions are different (Fig. 2B; Supplementary Fig. 3C). The FRET pairs used for SUL25 were Alexa 488/ Cy5 (MFD) or Atto 532/ Cy5 (TIRF), and for UL25 were Alexa 488/ Cy5 (MFD) or Cy3B/ Cy5 (TIRF).

Note that intermolecular recombination of linked-site substrates is also possible and could result in multiple products following alignment and strand exchange of the X and Y-arms in a variety of orientations. However, the intramolecular pathway should be favoured by the high local concentration of the linked partner site. The extent of intermolecular recombination is predicted to be low in single-molecule experiments due to picomolar substrate concentrations and lack of substrate movement due to surface immobilization.

**Annealing.** DNA oligonucleotides were synthesized and labelled by ATDBio (Oxford) using NHS-esters of Alexa 488, Atto 532, or Cy5. The linked-site substrates generally consisted of 5 oligonucleotides (Supplementary Fig. 3) which were annealed using a single-step process. Samples (20 µl) containing the oligonucleotides (5 µM of each) in a buffer (10 mM Tris (pH 7.5), 100 mM NaCl) were heated to 85 °C and cooled slowly in a PCR cycler at −0.1 °C per cycle until 4 °C was reached. Standard PAGE purification was used to purify the final product and ensure that the fully assembled linked substrate was used in the recombination assays.

**Ensemble measurements.** Unless stated otherwise, all experiments were performed at 37 °C. For an ensemble assay, 10% v/v resolvase, diluted in a 'resolvase dilution buffer' (RDB) containing 20 mM Tris-HCl (pH 7.5), 1 mM DTT, 0.1 mM EDTA, 1 M NaCl, 50% v/v glycerol was added to samples containing 20 nM of substrate (SUL25 or UL25) in reaction buffer (50 mM Tris-HCl (pH 8.2), 5 mM MgCl$_2$, 50 µg/ml poly(dI-dC)). Samples were mixed thoroughly by vortexing and briefly centrifuged. Small volumes (20 µl) were taken from the master mix at different time points and the reaction was stopped by the addition of a

loading buffer containing SDS and proteinase K (267 mM Tris-HCl, 267 mM boric acid, 0.3% w/v SDS, 3 mM DTT, 0.015% w/v bromophenol blue, 12% w/v Ficoll and 3 mg/ml proteinase K). Samples of ~20 µl were loaded onto an 8% w/v polyacrylamide gel (37.5:1 acrylamide: bis-acrylamide) containing 0.1% w/v SDS, and electrophoresis was performed at room temperature (for linear substrates) or 4 °C (linked-site substrates). Bands were visualised with a Typhoon fluorescence imager (GE Healthcare).

**Single-molecule measurements.** Buffers for use in single-molecule experiments were pre-prepared and syringe-filtered (0.20 µm syringe filter (Millex, Millipore)) to ensure high purity.

For TIRF microscopy experiments, the imaging buffer (buffer A), composed of 20 mM Tris-HCl (pH 8.0), 6% w/v D-(+)-glucose (Sigma Aldrich), was used to prepare solutions of DNA to be immobilized onto the functionalized chamber surface and also as a wash buffer immediately prior to imaging. Buffer B, used to dilute protein before addition to immobilized DNA, contained 20 mM Tris-HCl (pH 8.0), 6% w/v D-(+)-glucose (Sigma Aldrich), 50 mM NaCl. The synaptic complexes were prepared in the same way as described for ensemble measurements in buffer B prior to dilution and immobilization. The synaptic complexes were diluted to ~20–50 pM DNA concentration and immobilized. A concentration of 5–20 mM MgCl$_2$ was added optionally. 2 mg/mL glucose oxidase, 0.08 mg/mL glucose catalase (Sigma Aldrich)), and 1 mM Trolox ((+)−6-Hydroxy-2,5,7,8-tetramethylchromane-2-carboxylic acid (Sigma Aldrich)) were added to buffer B prior to imaging to reduce the rate of blinking and photobleaching of the dyes. See TIRF methods for further explanation.

For MFD measurements, the buffer contained 20 mM Tris, 50 mM NaCl (pH 7.5) and optionally 5–20 mM MgCl$_2$. Before diluting the DNA samples to the single-molecule level, vitamin C was added into the buffers (1 mM). Samples containing DNA only were typically prepared at 20 nM, incubated at 37 °C (unless stated otherwise) for 1 h, then diluted to ~4 pM before measuring. Samples containing resolvase were typically prepared at 20 nM substrate concentration and 1.2 µM resolvase concentration (with 10% v/v resolvase dilution buffer) and were incubated at 37 °C (unless stated otherwise). The recombination reaction was stopped after various time points by diluting to ~4 pM substrate concentration before measuring with MFD (see below).

**Protein expression, purification and labelling.** Tn3 NM resolvase and Sin Q115R expression and purification were as previously described[31,44].

The sequence encoding S185C NM resolvase mutant was synthesized as a gBlock (IDT) and inserted into the NM resolvase expression plasmid pFO102[28], replacing the NM resolvase reading frame. As well as the cysteine mutation to allow conjugation of a fluorophore label (see below), the mutant reading frame contained a C-terminal polyhistidine tag. Purification of S185C NM resolvase was carried out as described by Arnold[45] but with modifications made to the protocol to allow for the use of steps relating to the purification of hexahistidine-tagged resolvase as described by Olorunniji et al.[46] HPLC was performed using an AKTA purifier system.

Before labelling, the purified S185C NM resolvase was buffer-exchanged into conjugation buffer (20 mM sodium phosphate (pH 7.0), 1 M NaCl, 5 mM EDTA) by dialysis, before reduction by addition of aqueous TCEP (tris(2-carboxyethyl)phosphine) solution (pH 7.0) (Melford) to a concentration of 2 mM and incubation at room temperature. The maleimide-modified lyophilized Cy5 (Thermo Fisher) was dissolved in anhydrous DMF (Sigma Aldrich), the solution of which was added to the reduced protein at >20x molar excess and then left at 4 °C overnight. Excess dye was deactivated by addition of 2-mercaptoethanol (Melford) to a concentration of 100 mM. The mixture was centrifuged, then diluted with imidazole-containing binding buffer (IBind) (20 mM sodium phosphate (pH 7.0), 1 M NaCl, 50 mM Imidazole, 2 mM TCEP). Unbound dye was then removed by metal

affinity chromatography. Eluted protein fractions with fluorophore absorbance were pooled and dialysed into Refold buffer (20 mM Tris-HCl pH 7.5, 1 mM DTT, 0.1 mM EDTA, 2 M NaCl) for removal of imidazole before further dialysis against RDB, then stored at −20 °C.

**In-gel FRET**. Fluorescent bands on gels were visualized using a fluorescence imager (Typhoon FLA 9500). Gel bands were subject to the following excitation and emission protocols, referred to in the text as a 'FRET scan'. Since most substrates contained two fluorophores or more, with overlapping donor emission/ acceptor absorption spectra to facilitate FRET, steps had to be taken to avoid spectral crosstalk and ensure discrete detection, so two consecutive gel scans were taken. All fragments shown in gels featured in this work contained Alexa 488 and/or Cy5. Scanning the Alexa 488 and Cy5 FRET pair required the use of a bandpass filter optimized for Alexa 488 (BPB1(530DF20), Fujifilm). The laser line used for Alexa 488 excitation has a wavelength of 473 nm which lies slightly outwith Cy5 absorption; therefore, the use of the bandpass filter was precautionary, to completely prevent detection of any Cy5 emission. Two separate scans were conducted using $\lambda_{ex\,(Alexa\,488)} = 473$ nm, $\lambda_{em\,(Alexa\,488)} \sim 530$ nm (bandpass blue filter, Fujifilm) and $\lambda_{ex\,(Cy5)} = 635$ nm, $\lambda_{em(Cy5)} \geq 665$ nm (long pass red filter, Fujifilm). If the experiment focused on visualizing changes in FRET efficiency during the recombination assay, a third scan was performed using $\lambda_{ex\,(Alexa\,488)} = 473$ nm, and $\lambda_{em\,(Cy5)} > 665$ nm. Fluorescence intensities of the bands were determined in ImageQuant (GE Healthcare) by densitometry. Bands containing Alexa 488 fluorescence were given the colour green, those containing Cy5 were coloured red, and those exhibiting FRET were coloured blue. When all three scan images were overlaid, bands containing both fluorophores situated far apart (not within FRET range) appeared yellow, and those containing both fluorophores close together, causing FRET, appeared white.

**MFD**. Buffers were prepared with Tris (Sigma-Aldrich), Tris-HCl (Sigma-Aldrich), NaCl (Fluka) and $MgCl_2$ (Fluka) in ultrapure water (Direct Q3, Merck Millipore). Measurement buffers were cleaned using activated charcoal. MFD measurements were performed using a home-built system. The fluorescent donor molecule (Alexa 488) was excited by a linearly polarized laser (480 nm, 40 MHz, ~60 ps FWHM; Picoquant, Germany). The laser light was focused into the dilute solution of labelled molecules by a water immersion objective (UPLAPO 60×, NA = 1.2, Olympus, UK). The average laser power at the sample was ca. 180 µW. All measurements were recorded at 21 ± 1 °C. The data analysis used software written by the group of Prof. Claus Seidel (Heinrich Heine Universität, Düsseldorf). FRET efficiencies were measured from raw green and red signals and corrected for background and crosstalk.

In MFD experiments with Cy5-labelled S185C NM resolvase (for purification and labelling, see above), our aim was to observe FRET changes indicating movement of a fluorophore-labelled protein subunit relative to another fluorophore attached to the DNA substrate, which might provide further information on the mechanism. Attachment of the fluorophore at the resolvase C-terminus (S/C185) was expected to minimize any interference with binding, synapsis or recombination (Supplementary Fig. 8A). The Cy5-labelled resolvase was mixed with an excess of unlabelled Tn3 NM resolvase and added to a UL25 substrate, labelled at one site I with Alexa 488.

**TIRF**. The laser used as the excitation source was a 532 nm Stradus 532-40 (Vortran) laser (for Cy3B and Atto 532 excitation), and flow cells were placed on top of an inverted microscope (IX71, Olympus). Unless stated otherwise the laser power during excitation of a sample of immobilized molecules was ~25 µW. Fluorescence emission from a sample or bead slide was collected using an oil immersion objective (100×, NA = 1.49, Olympus, UK) and separated from scattered excitation via a 550 nm long-pass filter and a 532 nm band-reflect dichroic mirror (Chroma Technology Corp.). The fluorescence from sample molecules was

collimated and split into donor and acceptor wavelengths (DV2 Multichannel Imaging System, Photometrics) and simultaneously imaged onto a cooled EMCCD camera (Evolve, Photometrics). This provided two separate spatially identical images displaying donor and acceptor fluorescence. Experiments involving Sin Q115R were carried out at 37 °C unless stated otherwise. Experiments involving Tn3 NM were carried out at 21 ± 1 °C. The fluorescent sample concentration was accordingly adjusted at the beginning of the experiment such that approximately 200 spots were visible within the field of view, and .tif movies were recorded using Image Pro-Plus 7.0 software with an exposure time of 50 ms.

**Recording movies**. Linked site I synapses were immobilized to the cover slip surface using biotin-neutravidin interactions at an approximate concentration of 10 pM. The fluorescence intensities of the emission from the Atto 532 (donor) and Cy5 (acceptor) dyes were detected using an EMCCD camera and .tif movies were recorded by ImagePro-Plus 7.0 software using an exposure time of 50 ms.

**Processing movies**. Movies recorded as .tif files were gathered using the programme ImagePro-Plus 7.0 and analyzed using a commercially available programme called TwoTone 3.1. Images of bead slides which emitted strong fluorescence in both donor and acceptor channels were taken at the beginning of every TIRF experiment. These images were used to generate a mapping file in TwoTone to calibrate the donor and acceptor channels such that a spot observed in both channels is recognized as belonging to fluorescence from the same molecule. Once loaded onto the software, intensities threshold can be set for each movie in both channels to ensure that only the brightest spots (usually ~200) were analysed. Intensities of FRET pairs from several molecules recorded for every frame of the movie were recorded as a MATLAB array. Fluorescence vs. time trajectories were then visualized using MATLAB.

**Transition analysis using MASH-FRET**. All dynamic trajectories were analysed individually without stitching using software packages 'HaMMy'[33,46] and 'MASH-FRET' version 1.3.2[34], in a similar method to that described previously by Bianco et al.[35]. Unstitched time trajectories were used to produce transition density plots (TDPs) which were analyzed by fitting to $K(K-1)$ isotropic 2D Gaussians, where K is the number of FRET states. FRET values were derived from the Gaussian means and the associated errors from the average Gaussian sample standard deviations (SD) in MASH-FRET. The value of $K_{opt}$ was inferred via ML-BIC optimization of models comprising $K = 1$ to 10 states with 30 initializations. Transition rate coefficients were estimated using the 'state lifetimes' method described by Hadzic et al. [34]. Cumulative dwell time distributions were fit to single and biexponentials. To determine the rate coefficient of each transition, state lifetimes were multiplied by the weight of each corresponding cluster determined from the TDP analysis. An average value of the rate constant ($k_{av}$) was calculated by weighting each dwell time component ($k$) by the amplitudes (A): $k_{av} = \frac{1}{\left(\frac{A_1}{k_1}\right) + \left(\frac{A_2}{k_2}\right)}$. The error in dwell time (3 × SD) for each transition was determined using bootstrapped exponential fitting (BOBA-FRET)[35,47]. To determine the SD in the average dwell time, we separately calculated the SD in $\left(\frac{A_1}{k_1}\right)$, which we define as '$M$' and $\left(\frac{A_2}{k_2}\right)$, which we define as

'$N$', by using $\Delta M = \left(\frac{A_1}{k_1}\right) \times \sqrt{\left(\frac{\Delta A_1}{A_1}\right)^2 + \left(\frac{\Delta k_1}{k_1}\right)^2}$ and $\Delta N = \left(\frac{A_2}{k_2}\right) \times \sqrt{\left(\frac{\Delta A_2}{A_2}\right)^2 + \left(\frac{\Delta k_2}{k_2}\right)^2}$ where $\Delta A$ and $\Delta \tau$ are the SDs in amplitude and dwell time, respectively, from BOBA-FRET analysis. The overall SD in $k_{av}$ ($\Delta k_{av}$) was calculated as $\Delta k_{av} = \sqrt{M^2 + N^2}$.

**AV simulations.** Accessible volume (AV) simulations were performed as described by Kalinin et al.[48], using freely available software FPS version 1.1. We used pdb files from available crystal structures including the structure of a γδ resolvase tetramer in a synaptic complex with cleaved site I DNA (PDBid 1zr4)[9]; the structure of a γδ resolvase dimer bound to site I DNA (PDBid 1gdt)[7]; and a model created previously by Nollmann et al.[11] from two 1GDT structures to represent a low-resolution structure of the uncleaved Tn3 resolvase synapse. We used "three radii AV", a superposition of AV clouds calculated using three dye radii, which required different dimensions for each fluorophore used along with the number of linker atoms, width and length of the linker. We used dye parameters for nucleic acids labelled at a pyrimidine base with "C6-amino linker" and fluorophores Alexa Fluor 488 NHS ester, Atto 532 NHS ester and Sulfo-Cy5 NHS ester. Atom IDs used to define various attachment points were based on individual locations of the fluorophores for each substrate (which were slightly different for Tn3 and Sin substrates). The AV cloud generated could then be visualised in PyMOL. After the simulation was complete for both donor and acceptor fluorophores, the software was used to determine average FRET distances using the Forster radii for each chosen pair (all shown in Table 1).

### Reporting summary

Further information on research design is available in the Nature Portfolio Reporting Summary linked to this article.

## Data availability

The data sets generated and analysed during the current study are available from the University of Glasgow data repository Enlighten [https://doi.org/10.5525/gla.researchdata.1641].[49] Note that not all data are available to download directly from the repository due to file size but are available upon request. Source data are provided with this paper.

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

## Acknowledgements

We are very grateful to Prof. David Lilley, Dr. Anne-Cécile Déclais (University of Dundee) and members of their laboratory for help with our preliminary FRET experiments. This work was funded by a project grant from the Biotechnology and Biological Sciences Research Council (BB/R008493/1), a PhD studentship from the University of Glasgow College of Medical, Veterinary & Life Sciences (to G.M.C.), and a PhD studentship from the Wellcome Trust (to J-G.S.).

## Author contributions

S.W.M. and W.M.S. initiated the project, obtained funding, and designed the experimental programme along with J.-G.S., G.M.C. and M.R.B. Most of the experimental work reported in this manuscript was performed by G.M.C., J.-G.S. developed DNA substrate designs and methodologies, and carried out initial single-molecule FRET studies. G.M. performed the studies with fluorophore-labelled resolvase mutants. Analysis of the data was by G.M.C., W.M.S. and S.W.M. The manuscript was written by G.M.C., W.M.S. and S.W.M. with critical reading and advice from M.R.B. who also provided Sin Q115R protein used in the experimental studies.

## Competing interests

The authors declare no competing interests.
