## [Transparent Peer Review file · Nature Communications]

Direct observation of subunit rotation during DNA strand exchange by serine recombinases

Corresponding Author: Dr Steven Magennis

Version 0:

Reviewer comments:

Reviewer #1

(Remarks to the Author)

In this manuscript, Cadden and coworkers have succeeded in observing reaction intermediates in the serine recombinase reaction using solution and surface smFRET. Serine recombinases use a 'subunit rotation' mechanism to exchange strands between DNA substrates that involves large changes in quaternary structure and exposure of double strand breaks. The mechanism was dismissed as implausible when first proposed many years ago, but is now largely accepted based on a variety of experiments by several researchers. However, in nearly all cases, the subunit rotation part of the mechanism has been inferred rather than directly observed. Here, the SR reaction has been monitored using changes in FRET efficiency that occur as the DNA sites come together, are cleaved, undergo rotation, and are then re-ligated.

This is an important paper and I have no major concerns about how the experiments were designed and performed. While I fully support publication of this work, I do have some concerns about the way the paper has been written and some comments about the interpretation of the FRET efficiencies. These are not meant as criticisms of the work, but rather an attempt to make the paper more accessible to readers of NCOMM.

First, the organization of the paper. The Introduction is fine. The Results section, however, will be difficult to follow for a general reader who is interested in what was learned about serine recombinases because it is really not explained well until the middle of the Discussion and reference to Fig. 7. Much of the text in the Results could easily be moved to the Methods section, since it involves technical aspects of the substrates, the smFRET measurements, and processing of the data. This would allow a more concise presentation of results, perhaps using some of the Discussion text, that seems more appropriate for a Communication format paper. The Discussion is long, partly because it repeats the introduction and presents the results of the paper.

Second, the interpretation of FRET data in the context of a model. The authors could be correct in their assignment of structural intermediates (lnr, cnr, cr, lr) to the FRET efficiency gaussian components extracted from the TIRF measurements. The argument that base-pairing at the start and end of rotation could result in an energetic 'pause' seems reasonable. However, there could be alternative interpretations and I worry that the available crystal structures have overly biased what might be learned from the FRET experiments. The crystal structures need not all correspond to a high FRET intermediate and the highest FRET states need not correspond to a crystal structure. For Sin, a very broad distribution of FRET efficiencies is centered at $E=0.47$; this corresponds to neither the cnr or cr intermediate/structure, but rather lies in the middle of the E values calculated from accessible volumes based on the two structure models (0.31 and 0.66). Why could this not represent a freely rotating species, with the fit gaussian representing the distribution of $\langle R \rangle$ values during the 180 deg rotation?

Some minor suggestions:

p. 3: The 'axle' analogy doesn't seem right to me. An axle is a rod that facilitates rotation. Why not use the 'bearing' model that was already used? That seems like a better fit.

Fig 2c: Describing the inversion and deletion pathways as associated with 'parallel' and 'antiparallel' synapsis will not be helpful for most readers; it may even be confusing since the crossovers are symmetric in the substrates. These terms are probably not necessary here, since it is clear why you get one or the other depending on how the sites come together.

Fig. 4: The UL25 and SUL25 substrates recombine differently, with Tn3-catalyzed deletion accumulating over time and the Tn3 catalytic mutant showing an observable FRET signal. Since this doesn't happen with the Sin mutant, this implies some difference between the two site-I complexes (e.g., DNA-bending). This suggests that the Sin data may be more reliable, right?

p. 9: The experiments with resolvase labeled with Cy5 at the C-terminus don't really add anything to the results since the data are hard to interpret. A single sentence saying that this approach was explored might be enough.

p. 14: With respect to the differences between the rates determined here and those observed in magnetic tweezer experiments, I agree with the explanation, but some differences in rates could also be due to the artificial nature of the system. The mutations used to activate Sin and Tn3 on simple substrates may also affect the reaction rates.

Reviewer #2

(Remarks to the Author)

Referee comments:

Direct observation of subunit rotation during DNA strand exchange by serine recombinases
by Cadden et al.

The manuscript by Cadden et al. describes the observation of subunit rotation of serine recombinase enzymes during DNA strands exchange using single molecule FRET (smFRET). The authors use linked-site substrate design for two recombinases: Sin and Tn3. The linked-substrate design allows to follow synapsis and recombination by following the FRET efficiency and favors intramolecular recombination pathway. First the authors select the suitable labelling positions of the donor and acceptor fluorophores on DNA substrate based on the amount of recombination product and model the FRET distance using accessible volume (AV) simulations. Then the subunit rotation during recombination is studied in free solution by MFD and immobilized on a surface by TIRF microscopy. Analysis of the trajectories gives the rate constant for each transition state. Overall, the results are well presented and discussed.

Please find our comments below, structured into major and minor points.

Major comments

1) Why different donor-acceptor fluorophore pairs have been used for MFD and TIRF?

2) In Table S2, the authors say that the fluorophore labelling positions were selected based on the amount of recombination products. This amount was determined by an arbitrary scale based on visible detection. How is this quantified? Is +5 and -16 not chosen in spite of similar values because these are too far positions for FRET?

3) In the explanation of Figure 6C (page 11), the authors claim that the most frequent transitions occur between *cnr* and *lnr*; however, the data in Figure 6C and S11D indicate that the most frequent transitions occur between the states *cnr* and *cr*. On a related note, could the authors elaborate why the most frequent transitions occur between *cr* and *lr* in Tn3 but between *cnr* and *cr* in Sin (Figure 6C)?

4) The authors are requested to elaborate more on why the experimental E value for Sin *lnr* complex is higher than predicted via AV simulation in Table 1. This makes the E value for *lnr* higher than *cnr* which is different from the trend in Tn3. How to ensure that the 0.16 FRET E observed for Sin is actually for the *cnr* state and not the *lnr* state?

5) The Methods section for Transition analysis using MASH-FRET (page 21), says that cumulative dwell time distributions were fit to single-exponentials to access the state lifetime. But the biexponential model gives a better fit. The authors speculate (page 17) that this may be due to the asymmetry of the synapse as shown in Extended data Figure 5. However, the values of $k_j(1)$ and $k_j(2)$ for the two individual components in Table S4 are very different from the single exponential fit ($k_{jj'}$). Could the authors elaborate more on this? If it was due to asymmetry, one would expect at least one of the two values to match with the single exponential fit.

6) Why were the TIRF experiments for the Tn3 NM resolvase done at 21 °C but for Sin done at 37 °C? What can be expected from the difference in temperature? Would it be similar to the trend of faster recombination at 37 °C as seen for Sin in Figure S2? How would that affect the rate constants?

7) The authors claim that the rates of transition between the two states is similar for the two systems (page 14). Are the two systems here indicating Sin and Tn3? If yes, then the $k_{jj'}$ value in single exponential fit for the same transitions in Table S4 show a significant difference in some cases. For example, the *cnr* to *cr* transition for sin has $k_{jj'} = 0.32 \text{ s}^{-1}$ and for Tn3 has $k_{jj'} = 0.52 \text{ s}^{-1}$. The *lnr* to *cr* for Sin has $k_{jj'} = 0.75 \text{ s}^{-1}$ and for Tn3 has $k_{jj'} = 0.05 \text{ s}^{-1}$. What could be the origin of these differences?

8) The authors say that "direction of motion cannot be distinguished based on the steady-state FRET populations, because

the start and end states are identical, this question might be addressed by kinetic analysis" (Page 17). Why "might"? Is the direction of motion already addressed in this work? How? In the introduction/discussion, please discuss what is needed in the experiment so that $+180^\circ$ vs. -180° rotation directions can be distinguished. Are the here presented experiments capable of identifying the rotation direction?

9) Since very different molecular behaviors were observed in Fig. 5C,D: could the authors categorize and quantify the overall occurrence of these distinct behaviors? What do the individual histograms for each category look like? In addition, a sketch and discussion of the expected (based on the model) FRET E traces for inversion, deletion, DSB etc. would help the reader to make sense of the observed variety of traces. This topic is currently hard to follow.

10) For the FRET histograms in Fig. 5, a logical fitting approach would be to use the peaks in A, B) as constraints for peaks in C,D). Can the authors show and discuss such an analysis?

11) The problem with the transition density plots is that they are the result of a pre-imposed model used for HMM fitting. In particular, fitting noise adds defective points to the TDP. What are the authors confident to conclude from the TDP? Could they show TDPs per category requested in Point 9) above?

12) During HMM fitting of TIRF/FRET traces, were all HMM parameters kept free for each trace? What results are obtained if the FRET efficiencies are fixed to those fit in Fig. 5C,D when using the peaks in Fig. 5A,B as constraints (as described in point 10). This would help to reduce the degrees of freedom and, in doing so, lower the risk of fitting noise. What effect does such a constrained HMM fitting have on the reported biexponentiality of the kinetics (cumulative dwell time plots)?

Minor comments

1) In Figure S4, what does the small greenish circle indicate? One line in the caption linking the different conditions A, B, C, D, E to the bands observed in B would be helpful.

2) In Figure 3 and Figure 7 the data for Tn3 resolvase comes earlier than Sin resolvase; however, in most of the other figures have the Sin data first. Uniformity in the order would make it easier to follow.

3) In the Methods section for TIRF (page 20), the objective is described as "a 100 \times 1.49 numerical objective". Is this the same as 100X objective?

4) In the SI extended data Figure 4D and S11, the graph for relative number of transitions indicate the FRET E changes instead of the names of the states between which the transitions take place. This makes it difficult to grasp.

5) In SI Figure S12, the graphs would be easier to follow if the graphs mention Monoexponential fit and Biexponential fit instead of ExpDecay 1 and ExpDecay 2.

6) In SI Table S4, the table would be easier to read if "single-exponential" and "double-exponential" was mentioned at the top of the table next to MASH-FRET and Origin respectively.

7) In Figure 4A high FRET is indicated with pink, but in Figure S2, the high FRET is indicated as pink/white. Consistent colours would improve readability.

Reviewer #3

(Remarks to the Author)

Cadden et al present an analysis of the mechanism of site-specific recombination by small serine recombinases using single-molecule FRET approaches backed up by controls using gel-based biochemical techniques. The central design is to use FRET pairs on the DNA strands to follow the exchange during recombination to test a long-standing model of subunit rotation. Although "controversial" in so far as the model at first seemed counterintuitive, there is now a fairly large body of structural and mechanistic evidence that is in support of the model. What the authors seek in this paper is to provide "direct observation of subunit rotation". Given the title, what I was excitedly expecting what data that tracked the FRET efficiencies as modelled in Extended data Figure. 5 or use of DNA or protein variants that trapped such intermediates. However, as the authors note, the kinetics of rotation is likely very fast (on a diffusion timescale) so cannot be directly observed using these approaches. What they instead present are equilibrated FRET efficiencies that represent start and end states in the process. Their data does provide accurate transition times between these states, but it is not clear as presented that this does more than provide further solid supporting evidence (as opposed to a major breakthrough). The data here differs from some previous clever single molecule magnetic tweezers approaches in that there isn't a driving torque, so the rotation is likely stochastically driven by thermal motion alone. Hence the data here has value in that regard.

The key argument from their discussion to support their title is on page 15 where they state that, "Individual molecules show frequent transitions between the *cnr* and *cr* states over the ~25-second observation period (Fig. 5D, S9B), consistent with multiple rounds of 180° rotation (either left- or right-handed; see above). The data would also be consistent with larger rotations of 360° or more; any rapid rotation (with intermediate state dwell times shorter than our 50 ms frame rate) of an odd multiple of 180° would be indistinguishable from a single 180° rotation, and an even multiple of 180° rotations would be

FRET-silent (see Results section).” This data interpretation seems to be central, yet the most convincing examples are in the supplementary figures. Critically, it wasn’t clear to me why in this topology-neutral experiment that the $cnr \leftrightarrow cr$ transitions couldn’t be switching back and forth (between left- and right-hand rotations), a result that could be supported by other models as well as subunit rotation. The paper could be strengthened by counterarguments against the other models that are not explained anywhere in the text. An added complication is that the high FRET state (the ~260 degree rotation in Extended data Figure 5) has a value close to the lr state. The argument that the biexponential transition times hint at differences in rotations would need to be explored in more depth by examining the correlations within and between single molecules events to explore and rule out static or dynamic enzyme disorder.

It is difficult to easily envisage control or additional experiments that might add to the data. The authors already considered mismatching in the recombination sites and hint at “designing synapses that are biased to move in a specific direction”. Such controls may help to discount alternative explanations for their data. Apropos of the earlier evidence for subunit rotation, I cannot see how this data could be interpreted exclusively as such. Although beyond the scope of this paper and possible additional controls, a route that may be worthwhile would be to adapt the structures produced in <https://www.nature.com/articles/s41586-019-1397-7>. If such an origami rotor was attached in place of the poly dT linkers here, then subunit rotation would be significantly amplified. It may also be that the rotation drag of the DNA “blades” could slow rotation sufficiently.

Suggestions/corrections:

I think that in explaining the rationale, the modelling in Extended data Figure 5 should be discussed/presented alongside the more in-depth table in Figure 3. This should include presentation of the alternative models and how they might map onto the FRET states, and where different transitions would be expected between different models. The important point about why the $cnr \leftrightarrow cr$ transitions cannot be explained by back-and-forth motions needs considering. NB I didn’t understand the labelling of “forward” and “reverse” in Extended data Figure 5. Each 180-degree position is isoenergetic even though they are “recombinant” and “non-recombinant” and moving from left to right along the x-axis is the same direction of rotation. The reasoning for the big variation in modelled FRET between 90- and 360-degree states needs explanation by figures since the difference in the donor/acceptor geometries that gives rise to this is not self-evident to the casual reader. The consideration of the influence of the orientation factor needs to be explained.

I couldn’t find any mention of experimental N values for either single-molecule approach. For the TIRF FRET it would be worthwhile considering and comparing events between different DNAs and comparing transition times across sequential events, to see if this can explain the biexponential distribution. Averages and error bars in graphs are not defined.

In presenting; the FRET states in the graphs these could be labelled with the interpreted states (cnr etc); the relative number of transition bar graphs could be rotated 90 degrees so that the transition labels are not so hard to read.

P11, line 8, “the most frequent transitions, both in the presence and absence of Mg^{2+} , were between states cnr and lnr ” – think they mean cnr and cr ? This important part of the data could be shown more clearly with exemplar data/analysis. Figure S4 was very hard to follow and I am not a fan of labelling over gels. The gels could be separated so the labelling of fragments is easier to understand.

Version 1:

Reviewer comments:

Reviewer #1

(Remarks to the Author)

The authors have responded well to all of my comments and questions. I have no additional concerns.

Reviewer #2

(Remarks to the Author)

Cadden et al. have adequately addressed our questions and provided the necessary data. The following minor points should still be clarified.

1. In the caption of Fig 6C, the authors write that there are green bars whereas it is actually blue. Ideally, the temperature of the smFRET experiments should be mentioned in the caption of Fig 6, S15 and S17, similar to Fig S14.

2. Seeming inconsistency: while the authors write “a large of transitions were observed between the cnr and cr states” for both $Tn3$ and Sin , the highest relative number of transitions appear to be between cr and lnr (Fig 6C, for Sin) and between cr and lr (Fig S17D, for $Tn3$). What does this indicate?

3. In response to Comment 1, the authors write the excitation source for TIRF was green, however, they mention also a 488 nm excitation laser for TIRF in the Methods section. If true, the same fluorophores could have been used for TIRF and confocal detection after 488 nm. So why did the authors choose different fluorophores, nevertheless?

4. In Fig S17D the right side of the x-axis is cut off.

Reviewer #3

(Remarks to the Author)

I appreciate the changes that the authors have made to their manuscript to present their data and ideas more clearly. I am satisfied that they have adequately responded to my comments. It's a nice piece of work - well done.

Response to reviewers' comments for Nature Communications manuscript NCOMMS-24-21431-T

“Direct observation of subunit rotation during DNA strand exchange by serine recombinases”

by Gillian M. Cadden, Jan-Gero Schloetel, Grant McKenzie, Martin R. Boocock, Steven W. Magennis and W. Marshall Stark

We thank the reviewers for their helpful comments, which we have addressed in detail below and through changes to the main text and supplementary information.

Reviewer #1:

This reviewer found the work to be important and they have *“no major concerns about how the experiments were designed and performed.”* The reviewer also states that they *“fully support publication of this work.”*

This reviewer had some concerns about the structure of the paper, which we address below.

Comment 1) The Results section, however, will be difficult to follow for a general reader who is interested in what was learned about serine recombinases because it is really not explained well until the middle of the Discussion and reference to Fig. 7. Much of the text in the Results could easily be moved to the Methods section, since it involves technical aspects of the substrates, the smFRET measurements, and processing of the data. This would allow a more concise presentation of results, perhaps using some of the Discussion text, that seems more appropriate for a Communication format paper. The Discussion is long, partly because it repeats the introduction and presents the results of the paper.

Response: We have moved the more technical aspects of substrate design to the Methods section, so the results are more immediately accessible to the general audience. We have also moved technical details of the single-molecule analysis techniques to the Methods section. We have also removed any parts that may have been considered repetitive or unessential.

Comment 2) The authors could be correct in their assignment of structural intermediates (Inr, cnr, cr, Ir) to the FRET efficiency gaussian components extracted from the TIRF measurements. The argument that base-pairing at the start and end of rotation could result in an energetic ‘pause’ seems reasonable. However, there could be alternative interpretations and I worry that the available crystal structures have overly biased what might be learned from the FRET experiments. The crystal structures need not all correspond to a high FRET intermediate and the highest FRET states need not correspond to a crystal structure. For Sin, a very broad distribution of FRET efficiencies is centered at $E=0.47$; this corresponds to neither the cnr or cr intermediate/structure, but rather lies in the middle of the E values calculated from accessible volumes based on the two structure models (0.31 and 0.66). Why could this not represent a freely rotating species, with the fit gaussian representing the distribution of $\langle R \rangle$ values during the 180 deg rotation?

Response: Our assignments of the states are not based solely on interpretation of the FRET Gaussians because the breadth of experimental FRET distributions can obscure underlying complexity. Instead, our detailed analysis of the TDPs (explained in *Methods*), based on previous analysis methods verified by Bianco *et al.*, allows us to conclude that the data best fit a 4-state

model. One would expect a 3-state model to better fit the data if resolvase was freely rotating, as the reviewer suggests, but this is not what we find for Sin or Tn3. Although we agree that the crystal structures do not need to correspond to the stable conformations that exist in solution, the agreement is very good and these are structural intermediates that are likely to represent local minima in the energy landscape.

Importantly, through further experiments and analysis, we have now switched the assignment of two of the states for Sin (see response to Reviewer 2, comment 5).

Note that since we expect greater structural differences between Sin resolvase and the $\gamma\delta$ -resolvase crystal structure than for Tn3 resolvase, we do not expect perfect correspondence between the R values determined from smFRET and AV analysis.

Comment 3) p. 3: The 'axle' analogy doesn't seem right to me. An axle is a rod that facilitates rotation. Why not use the 'bearing' model that was already used? That seems like a better fit.

Response: We agree that the 'bearing' model description is more appropriate. We have changed the text accordingly.

Comment 4) Fig 2c: Describing the inversion and deletion pathways as associated with 'parallel' and 'antiparallel' synapsis will not be helpful for most readers; it may even be confusing since the crossovers are symmetric in the substrates. These terms are probably not necessary here, since it is clear why you get one or the other depending on how the sites come together.

Response: We have changed the terminology here, opting for 'inversion-ready' or 'deletion-ready' alignment in place of 'parallel' and 'anti-parallel', to simplify the description for the reader.

Comment 5) Fig. 4: The UL25 and SUL25 substrates recombine differently, with Tn3-catalyzed deletion accumulating over time and the Tn3 catalytic mutant showing an observable FRET signal. Since this doesn't happen with the Sin mutant, this implies some difference between the two site-I complexes (e.g., DNA-bending). This suggests that the Sin data may be more reliable, right?

Response: The substrates are designed to favour the 'inversion ready' alignment of sites in the synapse. To make a deletion product, the sites must synapse in the opposite alignment ('deletion-ready'). Deletion is a probable dead end if the product sites dissociate, whereas inversion is not- the sites can re-synapse and rotate again. One explanation for the difference in these observations is that Tn3 NM is faster (more rounds of recombination with dissociation of the sites), so it samples the 'minor' deletion-ready state more often. An alternative explanation is that Tn3 NM might more frequently make the deletion-ready alignment of sites, because of structural features of its synapse. Further analysis would be required for a full explanation.

Comment 6) p. 9: The experiments with resolvase labelled with Cy5 at the C-terminus don't really add anything to the results since the data are hard to interpret. A single sentence saying that this approach was explored might be enough.

Response: These experiments were primarily to further investigate the subunit rotation mechanism. We have explained this further in the text and reduced the level of detail about these experiments.

Comment 7) p. 14: With respect to the differences between the rates determined here and those observed in magnetic tweezer experiments, I agree with the explanation, but some differences in

rates could also be due to the artificial nature of the system. The mutations used to activate Sin and Tn3 on simple substrates may also affect the reaction rates.

Response: We agree with this comment. We have now made it clear in the text that the mutations to the proteins and use of oligonucleotide substrates are likely to mean that our observations are not precisely as would be observed for WT resolvase. We have added a comment to aid our explanation of rate differences between the different techniques.

Reviewer #2:

The reviewer found that that “*overall, the results are well presented and discussed*”.

Major comments

Comment 1) Why different donor-acceptor fluorophore pairs have been used for MFD and TIRF?

Response: This was due to our setup as our MFD is set up for blue excitation only (where we have used Alexa 488/ Cy5) as the FRET pair, but our excitation source for TIRF was green, and hence we switched to Atto 532 or Cy3B as our donor of choice.

Comment 2) In Table S2, the authors say that the fluorophore labelling positions were selected based on the amount of recombination products. This amount was determined by an arbitrary scale based on visible detection. How is this quantified? Is +5 and -16 not chosen in spite of similar values because these are too far positions for FRET?

Response: We thank the reviewer for highlighting that this has not been adequately explained. The choice was not solely based on the level of recombinant observed in these assays. We wanted the DA distance to be short initially (for a large change in FRET efficiency before and after recombination). +5 and -16 positions were not chosen over +5 and -6, as the DA distance was longer, so the change before and after recombination would not be large enough to detect clearly. We have offered a clearer explanation in the *Substrate Design* section of the main text.

Comment 3) In the explanation of Figure 6C (page 11), the authors claim that the most frequent transitions occur between *cnr* and *lnr*; however, the data in Figure 6C and S11D indicate that the most frequent transitions occur between the states *cnr* and *cr*.

Response: We thank the reviewer for noticing the *lnr/cr* discrepancy; this was a typo. We hypothesize that there may be a larger number of *cr* to *lr* transitions for Tn3 NM as we may record more molecules at a later stage of recombination during the measurement if the inversion reaction reaches equilibrium faster for Tn3 NM than for Sin Q115R (see ensemble data). Additionally, differences in the relative frequencies of the transitions between Tn3 NM and Sin Q115R may relate to the different reactivity of the proteins (e.g. frequency and persistence of the cleaved state will depend on the relative (and absolute) rates of cleavage and ligation; etc.).

Comment 4) On a related note, could the authors elaborate why the most frequent transitions occur between *cr* and *lr* in Tn3 but between *cnr* and *cr* in Sin (Figure 6C)?

Response: To rationalise the interpretation in response to comment 3 (above) further, we studied Sin Q115R recombination at room temperature. Since our ensemble data showed that recombination of linear substrates using Sin Q115R is much slower at room temperature than at 37 °C (see fig. S2), we expected to find more molecules exhibiting low-FRET transitions, which would indicate early-state conformational change due to cleavage. In support of this prediction, we found that at room temperature the highest number of transitions occurred for the Inr ↔ cnr states (Fig. S14).

Comment 5) The authors are requested to elaborate more on why the experimental E value for Sin Inr complex is higher than predicted via AV simulation in Table 1. This makes the E value for Inr higher than cnr which is different from the trend in Tn3. How to ensure that the 0.16 FRET E observed for Sin is actually for the cnr state and not the Inr state?

Response: We have re-analysed traces which show fluctuations between states 1 and 3 or states 2 and 3 separately from the other traces using HaMMMy and MASH, to better distinguish between the two 'low FRET' states which have similar E values. Upon analysis of the TDPs, there was an approximately equal number of transitions from 1 to 3 and 2 to 3. We also analysed traces showing transitions between multiple states separately from the other traces. This TDP shows a higher number of transitions between states 2 and 3. Taken together with the AV simulation data, we have instead assigned state 1 as the Inr state, and state 2 as the cnr state, as observed in data for Tn3 NM. Since the transitions between states 1 and 2 and states 1 and 3 could be less easily distinguished in the global analysis using MASH-FRET, this separate analysis may more accurately represent the data.

Comment 6) The Methods section for Transition analysis using MASH-FRET (page 21), says that cumulative dwell time distributions were fit to single-exponentials to access the state lifetime. But the biexponential model gives a better fit. The authors speculate (page 17) that this may be due to the asymmetry of the synapse as shown in Extended data Figure 5. However, the values of $k_j(1)$ and $k_j(2)$ for the two individual components in Table S4 are very different from the single exponential fit (k_{jj}). Could the authors elaborate more on this? If it was due to asymmetry, one would expect at least one of the two values to match with the single exponential fit.

Response: Since the single-exponential fit averages the entire kinetic behavior into one rate constant, while the biexponential fit distinguishes between two separate processes, the values of k from these two fitting approaches are not expected to match. Also note that the k_{jj} values are subject to weighting based on the number of transitions observed, which affects the final value. In contrast, no weighting has been added to the k_j values taken from the biexponential fit. We have updated the table to include average k values from biexponential decays, with and without adding weights from the TDPs (k_{avj} and k_{avjj} , Table S4).

Note also that in Figure 6F we compared the rate constants of single exponential decays fit to cumulative dwell time histograms. This has been updated such that instead we now compare an average of both rate constants from the biexponential fits. Since all data best fit to biexponential fits we find this is the best way to compare the data. This was not done in the previous version of the manuscript due to limitations in the analysis method, which we have overcome as outlined above.

Comment 7) Why were the TIRF experiments for the Tn3 NM resolvase done at 21 °C but for Sin done at 37 °C? What can be expected from the difference in temperature? Would it be similar to the trend

of faster recombination at 37 °C as seen for Sin in Figure S2? How would that affect the rate constants?

Response: We tested both Tn3 NM and Sin Q115R with linear substrates both at room temperature and at 37 °C in ensemble experiments. We found that the efficiency of recombination using Sin Q115R was greatly increased from RT to 37 °C (fig. S2). However, the same effect was not observed with Tn3 NM, and recombination was only slightly faster at 37 °C compared to RT (note that we have added these data to the Supplementary Information document, Fig. S3). Ultimately the decision was made to study the reaction with Tn3 NM and linked site I substrates at RT, since we observed very fast inversion rates in ensemble experiments (see Fig. 4F), and the goal was to observe conformational changes caused by rotation.

Additionally, we analysed and compared the rates of Sin Q115R recombination at 37 °C and at room temperature and have decided to include these data in the manuscript (see figure 6F, S14). We find that rotation is slower at room temperature compared with 37 °C.

Comment 8) The authors claim that the rates of transition between the two states is similar for the two systems (page 14). Are the two systems here indicating Sin and Tn3? If yes, then the k_{jj}' value in single exponential fit for the same transitions in Table S4 show a significant difference in some cases. For example, the c_{nr} to c_r transition for sin has $k_{jj}' = 0.32 \text{ s}^{-1}$ and for Tn3 has $k_{jj}' = 0.52 \text{ s}^{-1}$. The l_{nr} to c_r for Sin has $k_{jj}' = 0.75 \text{ s}^{-1}$ and for Tn3 has $k_{jj}' = 0.05 \text{ s}^{-1}$. What could be the origin of these differences?

Response: See response to comment 5 above. The new data comparing average rates from the biexponential fits give c_{nr} to c_r rate constants $k = 0.86 (0.12) \text{ s}^{-1}$ (Sin Q115R) and $k = 1.14 (0.29) \text{ s}^{-1}$ (Tn3 NM). These rate constants are similar, and from the ensemble analysis we expect the recombination rates of Tn3 NM to be higher than Sin Q115R. We find that the kinetic results for Sin and Tn3 are generally consistent. The k_{jj} values are weighted values which depend upon the number of transitions observed between states.

Comment 9) The authors say that “direction of motion cannot be distinguished based on the steady-state FRET populations, because the start and end states are identical, this question might be addressed by kinetic analysis” (Page 17). Why “might”? Is the direction of motion already addressed in this work? How? In the introduction/discussion, please discuss what is needed in the experiment so that +180° vs. -180° rotation directions can be distinguished. Are the here presented experiments capable of identifying the rotation direction?

Response: Using kinetic analysis we can identify kinetic heterogeneity. As discussed in the text, we may speculate that the structural heterogeneity implied by the biexponential rates is due to the asymmetry of the synapse (Figure S20 A-C, with some synapses preferentially rotating in one or the other direction. However, it is not possible using our data to assign one particular rate constant to a specific direction of rotation. This would require further analysis, possibly using substrates which could be biased to rotate in a single direction.

Comment 10) Since very different molecular behaviors were observed in Fig. 5C,D: could the authors categorize and quantify the overall occurrence of these distinct behaviors? What do the individual histograms for each category look like? In addition, a sketch and discussion of the expected

(based on the model) FRET E traces for inversion, deletion, DSB etc. would help the reader to make sense of the observed variety of traces. This topic is currently hard to follow.

Response: We have separated the various transitions observed in the data into different categories and re-analysed in HaMMY and MASH-FRET as discussed in response to comment 4. An explanation of the distinct behaviours from Sin and Tn3 resolvase recombination analysis is provided in these data, as well as the TDP plots and complete kinetic analysis. The similarities and differences in the behaviour of each protein are discussed throughout the text. We mention that “no large FRET increase is expected for the deletion pathway, as the fluorophores are on separate product sites”, and this is illustrated in Fig. 2, with the experimental E value for the ‘deletion-ready’ Tn3 NM synapse shown in Fig. S7C. Figs. 2 and 3 in the main text provide a clear prediction of expected changes in D/A distances and we have made clearer reference to these diagrams in the Fig. 5 legend.

Comment 11) For the FRET histograms in Fig. 5, a logical fitting approach would be to use the peaks in A, B) as constraints for peaks in C,D). Can the authors show and discuss such an analysis?

During HMM fitting of TIRF/FRET traces, were all HMM parameters kept free for each trace? What results are obtained if the FRET efficiencies are fixed to those fit in Fig. 5C,D when using the peaks in Fig. 5A,B as constraints (as described in point 10). This would help to reduce the degrees of freedom and, in doing so, lower the risk of fitting noise. What effect does such a constrained HMM fitting have on the reported bi-exponentiality of the kinetics (cumulative dwell time plots)?

Response: In principle, the use of FRET values from control samples as constraints is a good idea. However, it would not be helpful in this case because the samples in Fig. S13 A and B are for DNA alone, rather than for a synapse. The substrate traces and histograms in Fig. S13A and B serve to show that recombination results in only one FRET state, and that these are both easily distinguishable and show no dynamic FRET traces. These FRET states cannot be used as controls for the actual synapse.

Comment 12) The problem with the transition density plots is that they are the result of a pre-imposed model used for HMM fitting. In particular, fitting noise adds defective points to the TDP. What are the authors confident to conclude from the TDP? Could they show TDPs per category requested in Point 9) above?

Response: As discussed in the *Methods* section, trajectories were analysed with the software packages HaMMY and MASH-FRET using a method similar to that described previously by Bianco *et al.*(1). Unstitched time trajectories were used to produce transition density plots (TDPs), and these were analysed using a clustering algorithm, which fits the TDP to a mixture of K ($K - 1$) isotropic 2D Gaussians to obtain the optimal number of FRET states (K_{opt}). Using this method we can reliably obtain an understanding of the best model (number of FRET states) that fit the data, as well as compare the relative number of state-to-state transitions. As discussed in response to comment 4, we have now analysed individual TDPs for each category for both Sin and Tn3 resolvase data.

Minor comments

Comment 1) In Figure S4, what does the small greenish circle indicate? One line in the caption linking the different conditions A, B, C, D, E to the bands observed in B would be helpful.

Response: The green circle indicates fluorescein location (this information was missing from the legend and has been added). We have made it clearer what A, B, C, D, E relate to in the legend.

Comment 2) In Figure 3 and Figure 7 the data for Tn3 resolvase comes earlier than Sin resolvase; however, in most of the other figures have the Sin data first. Uniformity in the order would make it easier to follow.

Response: Both figures have been updated accordingly.

Comment 3) In the Methods section for TIRF (page 20), the objective is described as “a 100’ 1.49 numerical objective”. Is this the same as 100X objective?

Response: The ‘100’ is a typo and has been deleted.

Comment 4) In the SI extended data Figure 4D and S11, the graph for relative number of transitions indicate the FRET E changes instead of the names of the states between which the transitions take place. This makes it difficult to grasp.

Response: This has been changed such that the states are labelled as their assigned structure rather than E values, in agreement with the TDP in the main text.

Comment 5) In SI Figure S12, the graphs would be easier to follow if the graphs mention Monoexponential fit and Biexponential fit instead of ExpDecay 1 and ExpDecay 2.

Response: This has been changed accordingly.

Comment 6) In SI Table S4, the table would be easier to read if “single-exponential” and “double-exponential” was mentioned at the top of the table next to MASH-FRET and Origin respectively.

Response: This has been changed accordingly.

Comment 7) In Figure 4A high FRET is indicated with pink, but in Figure S2, the high FRET is indicated as pink/white. Consistent colours would improve readability.

Response: This has been changed accordingly.

Reviewer #3:

Comment 1) The key argument from their discussion to support their title is on page 15 where they state that, “Individual molecules show frequent transitions between the *cnr* and *cr* states over the ~25-second observation period (Fig. 5D, S9B), consistent with multiple rounds of 180° rotation (either left- or right-handed; see above). The data would also be consistent with larger rotations of 360° or more; any rapid rotation (with intermediate state dwell times shorter than our 50 ms frame rate) of an odd multiple of 180° would be indistinguishable from a single 180° rotation, and an even multiple of 180° rotations would be FRET-silent (see Results section).” This data interpretation seems to be central, yet the most convincing examples are in the supplementary figures.

Response: We appreciate this comment and have changed Fig. 5 to include more examples from the Sin Q115R analysis.

Comment 2) Critically, it wasn't clear to me why in this topology-neutral experiment that the $cnr \leftrightarrow cr$ transitions couldn't be switching back and forth (between left- and right-hand rotations), a result that could be supported by other models as well as subunit rotation. The paper could be strengthened by counterarguments against the other models that are not explained anywhere in the text.

Response: The subunit rotation mechanism for site-specific recombination by serine recombinases is now widely accepted within the scientific community. Most other models have been refuted previously. For example, the previously prevalent "end-swapping" model was incompatible with experiments in which resolvase subunits covalently cross-linked to their DNA sites retained recombination activity (2). Some of the reviewers' comments suggest that we did not make it clear enough that this paper is not about comparing models of site-specific recombination, but rather about direct observation of the mechanism at the single-molecule level. Therefore, we have altered the introduction in order to highlight the extensive experimental evidence and detailed structural analyses that have consistently supported this model over alternative hypotheses, such as the strand swapping or domain swapping models. These other models fail to account for the precise biochemical and biophysical observations that are elegantly explained by the subunit rotation mechanism, solidifying its status as the definitive framework for understanding serine recombinase-mediated recombination.

See comment 1 above.

Comment 3) The argument that the biexponential transition times hint at differences in rotations would need to be explored in more depth by examining the correlations within and between single molecules events to explore and rule out static or dynamic enzyme disorder.

Response: See response to comment 9 (reviewer 2).

Comment 5) It is difficult to easily envisage control or additional experiments that might add to the data. The authors already considered mismatching in the recombination sites and hint at "designing synapses that are biased to move in a specific direction". Such controls may help to discount alternative explanations for their data. Apropos of the earlier evidence for subunit rotation, I cannot see how this data could be interpreted exclusively as such. Although beyond the scope of this paper and possible additional controls, a route that may be worthwhile would be to adapt the structures produced in <https://www.nature.com/articles/s41586-019-1397-7>. If such an origami rotor was attached in place of the poly dT linkers here, then subunit rotation would be significantly amplified. It may also be that the rotation drag of the DNA "blades" could slow rotation sufficiently.

Response: We thank the referee for this interesting suggestion (use of a fluorophore-labelled DNA origami rotor), but it would be very difficult to correlate unambiguously movements of the rotor with structural changes of the synaptic complex.

Comment 6) I think that in explaining the rationale, the modelling in Extended data Figure 5 should be discussed/presented alongside the more in-depth table in Figure 3. This should include presentation of the alternative models and how they might map onto the FRET states, and where different transitions would be expected between different models. The important point about why the $cnr \leftrightarrow cr$ transitions cannot be explained by back-and-forth motions needs considering.

Response: See response to comment 2 above.

Comment 6) NB I didn't understand the labelling of "forward" and "reverse" in Extended data Figure 5. Each 180-degree position is isoenergetic even though they are "recombinant" and "non-recombinant" and moving from left to right along the x-axis is the same direction of rotation. The reasoning for the big variation in modelled FRET between 90- and 360-degree states needs explanation by figures since the difference in the donor/acceptor geometries that gives rise to this is not self-evident to the casual reader. The consideration of the influence of the orientation factor needs to be explained.

Response: We have altered the wording "forward" and "reverse" to "first rotation" and "second rotation", an explanation of which is also aided by the additional diagram in Figure S20. The implication of these data is that rotation in one direction would not follow the same FRET signature as a rotation in the opposite direction due to the due to the asymmetric positioning of the two fluorophores

Comment 7) I couldn't find any mention of experimental N values for either single-molecule approach. For the TIRF FRET it would be worthwhile considering and comparing events between different DNAs and comparing transition times across sequential events, to see if this can explain the biexponential distribution. Averages and error bars in graphs are not defined.

Response: The experimental N values for traces used for TIRF analysis are provided in the figures (shown on the histograms in Fig. 5, S13, S18). The number of bursts acquired for analysis of MFD data is provided in the 2D histograms. Error bars for all graphs are explained in the legends provided. For graphs showing analysis from gels, error bars represent the standard error of the mean taken from triplicate repeats of recombination assays. The error in each rate coefficient was determined using bootstrapped exponential fitting (BOBA-FRET) to quantify the variability of the rate constants across each sample of single molecules (explained further in *Methods* section).

As explained in our response to comment 9 (reviewer 2), the substrate would need to be re-designed to explore the meaning behind the biexponential distributions observed.

Comment 8) In presenting; the FRET states in the graphs these [TDPs] could be labelled with the interpreted states (cnr etc); the relative number of transition bar graphs could be rotated 90 degrees so that the transition labels are not so hard to read.

Response: The TDPs have been changed accordingly for better readability.

Comment 9) P11, line 8, "the most frequent transitions, both in the presence and absence of Mg²⁺, were between states cnr and lnr" – think they mean cnr and cr? This important part of the data could be shown more clearly with exemplar data/analysis.

Response: As explained above, this was a typo and has been changed. We have included examples of cnr to cr transition data, as well as examples of other types of transitions observed, in Fig. 5.

Comment 10) Figure S4 was very hard to follow and I am not a fan of labelling over gels. The gels could be separated so the labelling of fragments is easier to understand.

Response: The gels have now been separated such that the fragment labelling could be illustrated without covering the gels.

1. Bianco S., Hu, T., Henrich, O. & Magennis, S.W. Heterogeneous migration routes of DNA triplet repeat slip-outs. *Biophys. Rep.* **2**, 100070 (2022).
2. Mcllwraith MJ, Boocock MR, Stark WM. Site-specific recombination by Tn3 resolvase, photocrosslinked to its supercoiled DNA substrate. *Journal of Molecular Biology.* 1996;260(3):299-303.

Response to reviewer 2 comments for Nature Communications manuscript NCOMMS-24-21431-T

“Direct observation of subunit rotation during DNA strand exchange by serine recombinases”

by Gillian M. Cadden, Jan-Gero Schloetel, Grant McKenzie, Martin R. Boocock, Steven W. Magennis and W. Marshall Stark

1. In the caption of Fig 6C, the authors write that there are green bars whereas it is actually blue. Ideally, the temperature of the smFRET experiments should be mentioned in the caption of Fig 6, S15 and S17, similar to Fig S14.

Response: The caption has been changed to reflect the colour change to blue and the temperatures have been added to the legends of the figures specified.

2. Seeming inconsistency: while the authors write “a large of transitions were observed between the c_{nr} and c_r states” for both Tn3 and Sin, the highest relative number of transitions appear to be between c_r and l_{nr} (Fig 6C, for Sin) and between c_r and l_r (Fig S17D, for Tn3). What does this indicate?

Response: In the last paragraph of the results section we explained about the potential for intermediate states being missed due to fast rotation dynamics, relative to our camera's resolution, which is also in keeping with the observed heterogeneity in dynamics. We wrote:

"We also observed transitions in both systems that apparently skipped intermediate states, such as transitions from $c_{nr} \leftrightarrow l_r$ and $l_{nr} \leftrightarrow l_r$ (Fig. 6C, S14D, S15D, S17D). Skipped states can result from fast intermediate transitions with dwell times approaching the time of each 50 ms TIRF movie frame, causing them to be unresolved in HMM analysis (for example, an apparent direct transition from a hypothetical state 1 to 3 could be a composite of transitions from state 1 to 2 and then 2 to 3)."

3. In response to Comment 1, the authors write the excitation source for TIRF was green, however, they mention also a 488 nm excitation laser for TIRF in the Methods section. If true, the same fluorophores could have been used for TIRF and confocal detection after 488 nm. So why did the authors choose different fluorophores, nevertheless?

Response: This was a typo. The laser used for TIRF microscopy was green (532 nm). We apologise for any confusion caused.

4. In Fig S17D the right side of the x-axis is cut off.

Response: This has been fixed.